# FAILURE IS FEEDBACK: History-Aware Backtracking for Agentic Traversal in Multimodal Graphs

Joohyung Yun [1]   Doyup Lee [2]   Wook-Shin Han [3]

## Abstract

Open-domain multimodal document retrieval aims to retrieve specific components (paragraphs, tables, or images) from large and interconnected document corpora. Existing graph-based retrieval approaches typically rely on a uniform similarity metric that overlooks hop-specific semantics, and their rigid pre-defined plans hinder dynamic error correction. These limitations suggest that a retriever should adapt its reasoning to the evolving context and recover intelligently from dead ends. To address these needs, we propose FAILURE IS FEEDBACK, which casts subgraph retrieval as a *sequential decision process* and introduces two key innovations. (i) We introduce a *history-aware backtracking mechanism*; unlike standard backtracking that simply reverts the state, our approach piggybacks on the context of failed traversals, leveraging insights from previous failures. (ii) We implement an *economically-rational agentic workflow*. Unlike conventional agents with static strategies, our orchestrator employs a cost-aware traversal method to dynamically manage the trade-off between retrieval accuracy and inference costs, escalating to intensive LLM-based reasoning only when the prior failure justifies the additional computational investment. Extensive experiments show that FIF achieves state-of-the-art retrieval on three benchmarks. The project page is available at `failureisfeedback.github.io`.

## 1. Introduction

Searching the web has become a part of everyday life. This routine increasingly underpins multimodal retrieval-

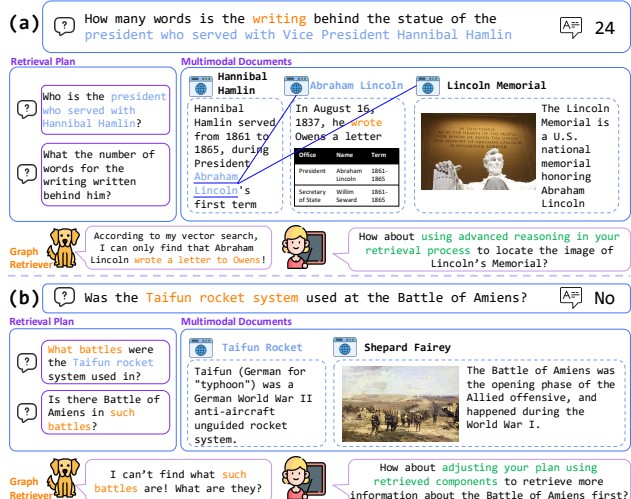

*Figure 1.* Motivating examples of multihop retrieval failures in existing graph retrieval approaches. (a) Vector-similarity-driven traversal follows a spurious cue. (b) Fixed retrieval plan produces an underspecified hop and fails to recover from a dead end.

augmented generation (RAG), where a model answers a user query by grounding its output in retrieved evidence (Park et al., 2025; 2026; Lee et al., 2025; Kang et al., 2025; Ling et al., 2025). In practice, much of this evidence lives in webpages or PDFs—multimodal documents with three salient characteristics: (i) each document is composed of multimodal *components* (paragraphs, tables, and images); (ii) the meaning of a component is often shaped by local document context (e.g., captions and surrounding components); and (iii) components are connected through explicit signals (hyperlinks, cross-references) as well as implicit signals (e.g., same-section adjacency). Moreover, documents themselves are linked via hyperlinks and citations, forming a large graph that users implicitly navigate while browsing. We refer to the resulting setting as *open-domain multimodal document retrieval* (OMDR): given a query, the system must return a small ranked set of relevant components from this large, noisy, and interlinked graph, often requiring multihop and multimodal exploration (Yun et al., 2025; Trivedi et al., 2023; Faysse et al., 2024).

Given these intricate characteristics of OMDR, representing a document collection as a graph has emerged as a powerful paradigm for capturing the multi-granularity and

[1]CSE, POSTECH, <jhyun@dblab.postech.ac.kr> [2]Director-Labs, <doyup@directorlabs.ai> [3]GSAI, POSTECH. Correspondence to: Wook-Shin Han <wshan@dblab.postech.ac.kr>.

*Proceedings of the 43rd International Conference on Machine Learning*, Seoul, South Korea. PMLR 306, 2026. Copyright 2026 by the author(s).

interconnectedness of multimodal evidence (Yun et al., 2025). It shows the pros of preserving the structural dependencies and navigational scaffolds inherent in webpages, which is required when navigating heterogeneous components. The most recent work introduces the *layered component graph*, which organizes components together with their constituent subcomponents (sentences, table rows, and image objects) (Yun et al., 2025). In this formulation, *navigational edges* encode relations among components (e.g., hyperlinks, same-section adjacency), while *hierarchical edges* connect each component to its subcomponents. By jointly modeling these edge types across layers, a retriever can traverse component-to-component paths for multihop exploration and move up/down the hierarchy to operate at the appropriate granularity.

While graph-based structures provide a rich representation of multimodal evidence, existing retrieval algorithms often struggle to fully exploit this potential due to operational rigidity. In particular, (a) traversal is typically driven by a single, hop-agnostic embedding-based scoring rule and (b) executed with a largely pre-specified procedure, limiting dynamic error correction. Figure 1 highlights these failures. In Figure 1(a), it follows a superficially related textual cue and retrieves an irrelevant snippet, failing to ground on the crucial visual evidence. In Figure 1(b), it issues an under-specified follow-up ("such battles") and gets stuck in a dead end, rather than adapting its trajectory after the failure. As a result, once the retriever follows a spurious edge or reaches a dead end, errors propagate across hops and degrade final retrieval quality (Trivedi et al., 2023; Asai et al., 2024). Moreover, these methods lack an explicit mechanism for deciding *when* expensive reasoning is warranted, leading to under-reasoning on ambiguous hops or over-spending computation across a trajectory.

To overcome these limitations, we argue that a retriever must evolve from a static path-follower into an adaptive decision-maker that navigates the graph through a sequential reasoning process. Concretely, OMDR is naturally stateful: as evidence accumulates, the information need shifts, and failures reveal which interpretations, routes, or strategies are unproductive. This suggests casting traversal as a *sequential decision process* over an evolving information state, where each step chooses (i) what to ask next (subquery), (ii) how to retrieve (tool/strategy), and (iii) where to move (edge type and granularity) conditioned on the current evidence. Achieving this requires addressing three coupled challenges. First, edge-following is not merely similarity matching; it often requires high-level reasoning to judge whether a candidate node will lead to the final answer under the current context. Second, the retriever must adapt to evolving context by refining hypotheses and subqueries, and by recovering from dead ends using failure signals rather than adhering to a fixed, pre-defined plan.

Third, it must balance accuracy and efficiency: while LLM-based reasoning can improve retrieval precision, it introduces substantial overhead, so the system must decide economically when to escalate from lightweight matching to intensive reasoning.

To address these needs, we propose FAILURE IS FEEDBACK (FIF). We formalize OMDR as a finite-horizon *information-state MDP*, where the state is a structured memory that records accumulated evidence together with the history of attempted subqueries, strategies, and explicit success/failure outcomes. This formulation turns graph traversal into an *economically-rational* agentic workflow: at each hop, an orchestrator dynamically decides *what to ask*, *how to retrieve*, and *where to move* given the current information state, rather than executing a rigid traversal recipe. To realize cost-sensitive control, FIF maintains a portfolio of strategies across an accuracy–efficiency spectrum, starting from low-cost vector matching and escalating to higher-cost LLM reasoning only when a hop is ambiguous or an attempt fails. Finally, to make multihop navigation resilient in noisy open-domain graphs, FIF introduces *history-aware backtracking*: unlike standard backtracking that simply reverts the state, our approach piggybacks on failure traces to re-anchor the search to a more promising prior context, revise subsequent subqueries, and avoid repeating previously failed routing patterns.

In summary, we make three primary contributions:

1. We formulate the OMDR problem as a sequential decision process with economic rationality. We redefine OMDR as an information-state MDP, operationalizing it through an LLM-enabled agentic workflow that treats retrieval strategy as a dynamic choice.

2. We propose dynamic cost-aware strategy escalation. We introduce a novel mechanism that maintains a portfolio of strategies across an accuracy-efficiency spectrum. Our orchestrator avoids over-reasoning by starting with low-cost vector matching and only escalating to high-cost LLM reasoning when a hop is identified as ambiguous or follows a recorded failure.

3. We propose history-aware backtracking for resilient navigation, which converts failed traversals into constructive feedback. By piggybacking on failure traces, the orchestrator re-anchors its search to prior contexts while revising its subqueries and escalating its strategy, enhancing both robustness and efficiency.

## 2. Related Work
### 2.1. Multimodal Retrieval Methods

Early multimodal retrievers were largely *TextRAG*-style: they transformed multimodal components into textual surrogates via OCR, captioning, or serialization, enabling mature text retrieval pipelines but inevitably discarding vision-

specific cues (Yang et al., 2023; Yu et al., 2023; Luo et al., 2023a). More recently, *VisRAG*-style pipelines unify modalities by rasterizing documents into page- or region-level screenshots and embedding all content in a single *visual* space (Yu et al., 2024; Faysse et al., 2024; Cho et al., 2025). However, they suffer from two key limitations: (i) *fixed granularity*, where large screenshots dilute query-relevant signals with irrelevant context, and (ii) *limited multihop reasoning*, treating pages independently without exploiting structural links (Chen et al., 2024; Zhong et al., 2025).

Closest to our work, LILAC represents a multimodal document corpus as a layered graph structure and performs structure-aware retrieval through vector-embedding-based graph traversal (Yun et al., 2025). It builds a component graph linking coarse nodes (paragraphs, tables, images) and fine-grained subcomponents, using edges to represent both hierarchical containment and navigational relations. At query time, LILAC performs an edge-wise beam search driven by late-interaction scores between subqueries and nodes. While effective, this pipeline operates under a rigid, pre-defined plan: it relies on a uniform similarity metric for traversal, overlooking hop-specific semantics, and executes a linear expansion strategy.

## 2.2. Graph Retrieval Methods

Graph-based retrieval has been extensively studied in knowledge graph QA, where systems traverse graphs curated with *typed* and *semantically meaningful* relations (Asai et al., 2019; Fang et al., 2020; Sun et al., 2023; Luo et al., 2023b; Xu et al., 2024). However, multimodal *document* graphs differ fundamentally: edges are primarily *navigational* (e.g., hyperlinks) rather than semantic predicates. Consequently, a retriever cannot treat traversal as simple path-finding over valid facts; it must perform online interpretation to resolve what a navigational link implies for the current query context (Yun et al., 2025). Thus, KG methods struggle with the adaptive capabilities to navigate large-scale, non-edge-labeled graphs where semantic resolution is needed (Yang et al., 2023; Hu et al., 2025).

## 2.3. Agentic Retrieval Methods

A growing line of work treats retrieval as an iterative decision process interleaved with reasoning, rather than a single-shot nearest-neighbor lookup. IRCoT shows that multi-step questions benefit from repeatedly generating intermediate sub-questions and retrieving evidence for each step (Trivedi et al., 2023). REACT formalizes a general reasoning-and-acting loop, motivating RAG controllers that plan retrieval actions based on intermediate observations (Yao et al., 2022). MOLORAG (Wu et al., 2025c) is a recent subgraph retrieval pipeline that scores unlabeled edges with an LLM under a fixed traversal strategy.

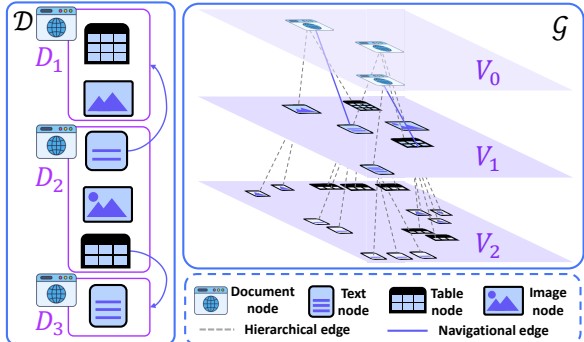

*Figure 2.* Visualization of an example corpus $\mathcal{D}$ and its corresponding layered component graph $\mathcal{G}$.

Despite this, most operate on *flat* indices, lacking the structural awareness to navigate inter-document links. Furthermore, their error correction is typically limited to query rewriting rather than history-aware backtracking. While systems like DOC-REACT (Wu et al., 2025b) and MARA (Wu et al., 2025a) apply agents to multimodal documents, they target single-document or small-scale contexts. They do not address the open-domain challenge of traversing vast, interconnected graphs where utilizing failure feedback is critical for routing optimization.

# 3. Multimodal Document Retrieval

We study *open-domain multimodal document retrieval* to find relevant components from a large multimodal corpus on a natural language query. In this study, we follow the setup of graph-based retrieval approaches (Yun et al., 2025), which have shown promising performances over existing naive approaches.

## 3.1. Problem Setting

**Corpus and components.** For the source of retrieval, a corpus $\mathcal{D} = \{D_1, \ldots, D_{k_{doc}}\}$ is a set of documents. Each document $D_j$ includes an ordered list of multi-modal components $D_j = [C_{j,1}, \ldots, C_{j,k_j}]$, where the global component pool is defined as $\mathcal{C} = \bigcup_{j=1}^{k_{doc}} \{C_{j,1}, \ldots, C_{j,k_j}\}$. The modality of each component $C \in \mathcal{C}$ can be a paragraph $P$, a table $T$, or an image $I$ as shown in Figure 2.

**Navigational links.** We assume a link signal $\mathcal{L}$ capturing navigational associations (e.g., hyperlinks and cross-document pointers), modeled as $\mathcal{L} : \mathcal{C} \to \mathcal{D}$.

**Multimodal retrieval task.** Given $Q$, $\mathcal{D}$, and $\mathcal{L}$, the retriever ranks components in $\mathcal{C}$ and returns $\mathcal{C}_R = [C_{R_1}, \ldots, C_{R_n}]$. Let $\mathcal{C}_{gt}(Q) = \{C_{gt_1}, \ldots, C_{gt_r}\}$ be the ground-truth relevant set; the goal is to rank its elements in $\mathcal{R}$, ideally near the top.

## 3.2. Layered Component Graph

We incorporate a layered component graph (Yun et al., 2025) to support an effective *coarse-to-fine* retrieval across

documents and their components. To efficiently retrieve relevant documents and evidence for a given query, open-domain retrieval requires to plan both which documents to visit and which components to read. Thus, we adopt the three-layered component graph to effectively represent documents, components, subcomponents, and their complex relations. Figure 2 shows an example graph $\mathcal{G}$.

**Nodes and Layers.** Let $\mathcal{G} = (\mathcal{V}, \mathcal{E})$ denote the graph. Using the definitions from Section 3, we construct a three-layered hierarchy $\mathcal{V} = V_0 \cup V_1 \cup V_2$:

- *Layer 0 (Documents) $V_0$*
- *Layer 1 (Components) $V_1$*: paragraphs/tables/images
- *Layer 2 (Subcomponents) $V_2$*: sentences, table rows, or visual objects

For nodes in $V_1 \cup V_2$, we save the raw multimodal content. Each $V_0$ node represents a document and serves as the anchor unit for document-level hops, organizing the document's components (via hierarchical edges) and its outgoing navigational links to other documents.

Different from LILAC (Yun et al., 2025), which uses two layers only with components and subcomponents, we add an explicit *Document Layer* with one node per document, which the retriever uses to shortlist documents and to expand to navigationally-linked neighbors. Adding document layer enables early pruning before descending to fine-grained evidence.

**Hierarchical Edges.** These edges represent the "contains" relationship, allowing the agent to drill down from coarse to fine granularity.

- Edges $(D_j, C_{j,i})$ for all components $C_{j,i} \in D_j$.
- Edges linking a component to its extracted subcomponents.

**Navigational Edges.** These edges capture explicit navigational paths across the corpus, allowing the retriever to transition between different document contexts based on the link signal $\mathcal{L}$. Using the link signal $\mathcal{L}$, we generate an edge $(C, D_k)$ if $\mathcal{L}(C) = D_k$.

Figure 2 illustrates the two edge types: dotted lines express hierarchical edges, and blue lines express navigational edges.

## 4. FAILURE IS FEEDBACK

We propose FAILURE IS FEEDBACK (FiF), an LLM-driven agentic retriever that traverses the layered component graph $\mathcal{G}$. Instead of executing a static, pre-defined traversal plan, FiF formulates retrieval as a *sequential decision process*: an Orchestrator iteratively chooses to (i) TRAVERSE from selected anchors under a strategy that explicitly trades off accuracy and cost, (ii) PLAN by revising or expanding subqueries as the information need evolves, or (iii) STOP and invoke a final RERANKER over all accumulated candi-

dates. A structured traversal memory records retrieved evidence together with explicit success/failure outcomes, operationalizing our core principle that *failure is feedback*: the Orchestrator escalates to stronger (but costlier) reasoning for traversal when lightweight hops are ambiguous or fail, and performs history-aware backtracking by re-anchoring to more promising prior contexts while avoiding previously failed routing patterns. We start by formalizing the sequential decision process, then explain the details for each agent that comprise the process.

### 4.1. Retrieval as a Sequential Decision Process

We formulate open-domain multimodal retrieval as a *sequential decision process* over the layered component graph $\mathcal{G}$. Given a query $Q$, the agent iteratively traverses $\mathcal{G}$ to output an ordered list of relevant components. We formalize the process as $\langle \mathcal{S}, \mathcal{A}, \mathcal{T} \rangle$, where the state is the agent's information state $s_t$; executing $a_t$ yields observation $o_t$, which is integrated into $s_{t+1}$ by the transition.

**State ($\mathcal{S}$).** We represent the information state as a structured *memory* $s_t = M_t$. $M_t$ is basically a trajectory log capturing decisions and outcomes: $M_t = (Q, \mathcal{Q}_t, H_t)$.

- *Original Query ($Q$):* The user's initial input.
- *Subquery List ($\mathcal{Q}_t$):* a list of decomposed query serving as a retrieval plan.
- *Action History ($H_t$):* an ordered sequence $[h_1, \ldots, h_t]$. Each record $h_k$ serves as a log of each action.

**Action ($\mathcal{A}$).** An action $a_t \in \mathcal{A}$ is a structured tool call selected by the orchestrator given $s_t$:

- TRAVERSE($\mathcal{D}_{anc}, q_t, \tau_t$) executes one retrieval hop for subquery $q_t$ with strategy $\tau_t$ from an anchor document set $\mathcal{D}_{anc}$.
- PLAN($M_t$) generates updated subqueries needed to solve $Q$, consulting both $\mathcal{Q}_t$ and $H_t$.
- STOP terminates the overall process. It applies a final RERANK module to all components stored in memory $M_t$ and returns a ranked list $\mathcal{C}_R$ based on their relevance to the original query $Q$.

**Observation ($o_t$).** $o_t$ contains the *new* outputs produced by the action at step $t$. For TRAVERSE, the observation is the traversed documents/components and if the traversal was successful or not. For PLAN, the observation is the updated list of subqueries.

**Transition ($\mathcal{T}$).** The transition process updates the state $s_t$ using the observation $o_t$ to generate the next state $s_{t+1}$. Specifically, it appends the observation to the action history $H_t$. It appends the newly generated subqueries to $\mathcal{Q}_t$.

### 4.2. Action Design

In this subsection, we detail the LLM-powered agents that implement each action in our sequential decision process

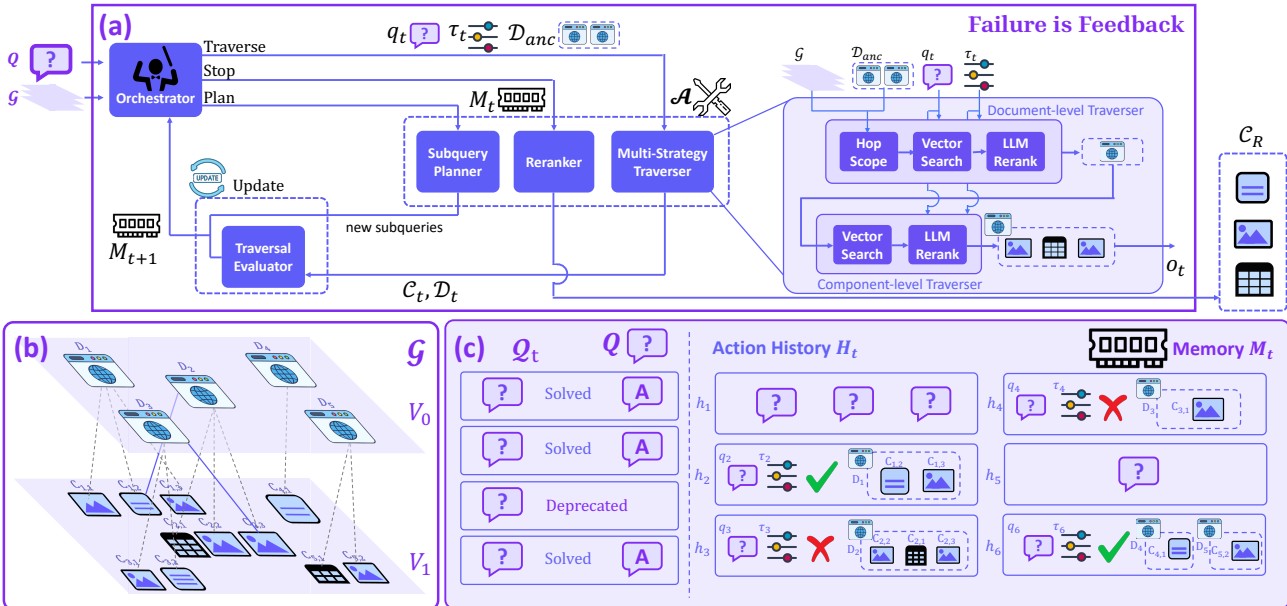

*Figure 3.* Overview of FAILURE IS FEEDBACK. (a) High-level orchestration loop and action interactions, labeled using the notation from the problem formulation (§ 4.1). (b) An example layered component graph $\mathcal{G}$. Note that only the document and component nodes are shown for brevity. (c) An example memory $M_t$ of a traversal over $\mathcal{G}$.

defined in § 4.1. Crucially, FIF's actions are designed to make two principles *operational*: (i) *economically-rational control* via a portfolio of traversal strategies that can be escalated on demand, and (ii) *failure-as-feedback* via explicit success/failure signals and history-aware re-anchoring for robust multihop navigation in noisy document graphs.

**(1) Multi-strategy Traverser.** It executes the TRAVERSE($\mathcal{D}_{anc}, q_t, \tau_t$) action on the layered graph $\mathcal{G}$, aiming to find subquery $q_t$-relevant components and documents, anchoring from the documents $\mathcal{D}_{anc}$. At the first hop ($t = 0$), $\mathcal{D}_{anc}$ is initialized to $\emptyset$ and the Traverser falls back to a corpus-wide global hop; for subsequent hops, $\mathcal{D}_{anc}$ is determined by the Orchestrator based on $M_t$. It consists of two stages: (i) a *document-level traverser* that selects a small set of candidate documents, and (ii) a *component-level traverser* that identifies $q_t$-relevant components among the candidates by searching their children nodes.

Crucially, the Traverser is a *configurable strategy engine*: the strategy tuple $\tau_t$ specifies retrieval behaviors that trade off accuracy and efficiency. **Strategy tuple.** $\tau_t$ controls traversal along three dimensions:

- *Hop Scope (Global vs. Local). Local Hop* restricts candidate documents to the direct neighbors of the anchor document set $\mathcal{D}_{anc}$ via the children's navigation edges; *Global Hop* considers the full document corpus to escape local neighborhoods when evidence is dispersed. It scores all corpus documents by max-pooling over their components and selects the most query-relevant

document set.

- *Vector Scoring Granularity.* It configures the layer at which vector similarity is computed. Intuitively, coarse-grained subqueries are often best matched at the component level, while fine-grained subqueries benefit from scoring at the subcomponent level. Both the document-level and component-level traversers use the granularities of $g \in \{1, 2\}$. Let $\text{Desc}_g(v)$ be descendants of $v$ at Layer $V_g$ (including $v$ if already in $V_g$). We denote by $x(\cdot)$ the multimodal encoder used for retrieval scoring, which produces an embedding for both a node $u$ and a query $q$. We score a node $v$ by

$$\text{SCORE}_{\text{vec}}(q_t, v; g) = \max_{u \in \text{Desc}_g(v)} sim(x(q_t), x(u))$$

- *LLM-Reasoning.* This option specifies whether to use LLM-based reasoning to accurately rerank components with $q_t$ beyond vector scores, supporting modes of {None, Component-only, Both}. To control LLM inference cost, we pre-filter top-$k$ candidates by $\text{SCORE}_{\text{vec}}$ and then pass their contents to the LLM for reranking.

The traversal outputs a document set $\mathcal{D}_t$ and a component set $\mathcal{C}_t$, which are recorded in the action history $H_t$.

**(2) Subquery Planner.** The Planner implements PLAN($M_t$) and generates new subqueries to address the remaining information needs for answering $Q$, conditioning on the current subquery statuses $\mathcal{Q}_t$ and the retrieved evidence stored in $H_t$. Invoking PLAN allows the agent to revise its plan using the newly acquired evidence

in the cases where initial subquery list is suboptimal. Newly generated subqueries are appended to the subquery list $\mathcal{Q}$ in memory $M_t$; importantly, we keep earlier subqueries rather than overwriting them, so the system retains a trace of what was tried and where the plan went off track.

**(3) Traversal Evaluator.** The Evaluator produces part of the observation $o_t$ of TRAVERSE by assessing whether the traversal outcome $\mathcal{C}_t$ is useful for the target subquery $q_t$. It provides two outputs. First, it judges whether the retrieved components can answer $q_t$; this success/failure signal is stored in the action history $H_t$ together with the traversal results. Second, it checks whether the retrieved content can resolve any remaining items in the subquery list $\mathcal{Q}_t$, and if so, extracts tentative answers and updates $\mathcal{Q}_t$ accordingly. These logged outcomes later guide backtracking decisions, helping the Orchestrator distinguish promising contexts from unproductive ones.

**(4) Reranker.** The Reranker is invoked by STOP to produce the final ranked list of components $\mathcal{R}$ from the accumulated memory $M_t$. Concretely, it aggregates all candidate components stored in memory and assigns each a final relevance score with respect to the *original* query $Q$, then returns $\mathcal{C}_R$ by selecting the top-$k$ components.

### 4.3. Orchestrator

The Orchestrator is the central LLM-driven controller that implements the policy over actions, selecting $a_t$ given the current information state $M_t$. At each iteration, it chooses among TRAVERSE, PLAN, and STOP to maximize retrieval accuracy while remaining efficient, treating *failure as feedback.*

Beyond selecting actions, the FIF Orchestrator actively *manages plan errors.* Rather than using $q_t$ as a direct copy from the subquery list, it treats the list as a scaffold and synthesizes a task-specific target by rewriting, refining, or composing subqueries based on the current evidence and unresolved constraints. In particular, it can use newly retrieved components to resolve missing information from earlier subqueries, or fuse multiple planned subqueries into a single sharper retrieval objective. When the subquery list becomes clearly unreliable (e.g., overly underspecified, drifting, or repeatedly unproductive), the Orchestrator can invoke PLAN at any time to regenerate a better set of subqueries conditioned on $M_t$, while preserving prior subqueries as a trace of what was attempted.

The Orchestrator also enables *history-aware backtracking via re-anchoring*: instead of always hopping from the most recent set of documents, it can resume from an earlier successful context recorded in $H_t$, turning failure traces into an actionable signal rather than treating dead ends as terminal states. Concretely, the Orchestrator carries out re-anchoring as a single joint update along three axes: (1) *revise* the sub-

query $q_t$ to incorporate constraints extracted from the failure trace; (2) *re-anchor* the anchor document set $\mathcal{D}_{anc}$ to the most-promising prior successful step found by scanning $H_t$; and (3) *escalate* the strategy tuple $\tau_t$ to avoid repeating the failed traversal behavior.

This jointly-decided update helps multihop traversal in two complementary ways. *(i) Correcting a misled anchor.* When the current $\mathcal{D}_{anc}$ has drifted into an unproductive region of the graph, re-anchoring snaps the search back to a prior context whose retrieved evidence already aligns with the revised subquery. *(ii) Strengthening per-hop reasoning when the prior tuple was too light.* When the previous attempt failed under a lightweight strategy tuple (e.g., vector-only matching without LLM reranking), the orchestrator could escalate $\tau_t$ to a more reasoning-intensive configuration.

We further codify a small set of prompt-encoded rules that govern when and how the three-axis update is performed. These rules concentrate the LLM's reasoning capacity on contextual judgment, while pinning mechanical decisions to predictable predicates; the full set is released with our official code. In principle, an LLM-driven Orchestrator could take all of these decisions on its own, and we expect this to become fully feasible as future LLMs deliver more efficient high-level reasoning; we therefore view the present rule set as a practical interim choice rather than a fundamental design commitment.

*When to backtrack and how to escalate the strategy tuple.* The Orchestrator initiates the three-axis update only when (a) $\geq 2$ consecutive failed traversals on related subqueries appear in $H_t$, or (b) the current $q_t$ has already been attempted under the most thorough strategy tuple—LLM reasoning at both the document- and component-selection stages—and still failed; otherwise it stays on the current anchor and revises only $\tau_t$ in place. When the update does fire, strategy tuples are advanced one position along a pre-defined cost ordering—from vector-only matching, up to LLM-reranked traversal at both the document- and component-level stages—so that each retry spends the smallest compute increment plausibly sufficient to resolve the failure rather than jumping to an arbitrary tuple.

*Local versus global scope escalation.* When deciding hop scope, the Orchestrator conditions on the navigational neighbor documents of the current anchor (surfaced from $\mathcal{G}$): it issues a Local Hop when these neighbors plausibly contain the missing evidence, and a Global Hop otherwise. The orchestrator widens the hop to global scope when the current $q_t$ has no viable anchor in $H_t$. This happens when either no prior successful step is relevant or all candidate anchors are failure-associated. This reflects an inductive bias that absent evidence cannot be recovered by stronger reasoning in the same neighborhood .

*Table 1.* Retrieval accuracy (Recall and MRR) of FiF and its competitors on three benchmarks. R@k, M@k indicates recall at $k$ and MRR at $k$, respectively. The best score in each column is in **bold**, and the second-best is underlined.

| Algorithm | MultimodalQADoc | | | | | MMCoQADoc | | | | | WebQADoc | | | | |
|---|---|---|---|---|---|---|---|---|---|---|---|---|---|---|---|
| | R@1 | R@2 | R@5 | R@10 | M@10 | R@1 | R@2 | R@5 | R@10 | M@10 | R@1 | R@2 | R@5 | R@10 | M@10 |
| NV-Embed-v2 | 26.42 | 37.63 | 52.08 | 61.45 | 68.13 | 18.22 | 28.46 | 40.22 | 48.19 | 41.97 | 22.18 | 31.38 | 39.80 | 51.19 | 55.81 |
| VisRAG | 34.19 | 42.12 | 53.38 | 56.91 | 57.88 | 19.57 | 27.53 | 33.59 | 37.31 | 30.01 | 25.58 | 41.46 | 46.10 | 48.75 | 50.60 |
| ColPali | 38.38 | 52.61 | 61.73 | 63.95 | 67.65 | 31.11 | 41.97 | 46.89 | 51.27 | 43.13 | 35.21 | 46.90 | 53.37 | 57.78 | 56.90 |
| LiLaC | 33.59 | 50.59 | 65.13 | 72.23 | 79.12 | 25.25 | 38.13 | 51.02 | 60.86 | 53.36 | 32.67 | 45.35 | 57.77 | 64.23 | 77.87 |
| IRCoT | 39.56 | 55.09 | 69.87 | 74.97 | 82.24 | 43.03 | 53.69 | 62.70 | 64.62 | 62.84 | 40.85 | 56.73 | 68.78 | 73.72 | 83.15 |
| MoLoRAG | 39.91 | 56.58 | 70.33 | 77.82 | 83.54 | 43.35 | 54.55 | 64.18 | 67.18 | 63.18 | 39.18 | 54.92 | 66.18 | 71.15 | 82.32 |
| FiF | **42.17** | **57.58** | **74.88** | **85.82** | **86.82** | **46.31** | **58.40** | **69.47** | **75.17** | **74.88** | **43.91** | **61.78** | **74.83** | **80.79** | **87.77** |

*Table 2.* End-to-end QA accuracy (EM and F1) of FiF and its competitors for the three benchmarks. The best score in each column is in **bold**.

| Algorithm | MultimodalQADoc | | MMCoQADoc | | WebQADoc | |
|---|---|---|---|---|---|---|
| | EM | F1 | EM | F1 | EM | F1 |
| NVEmbed-v2 | 53.80 | 61.15 | 39.20 | 47.73 | 51.82 | 59.88 |
| VisRAG | 51.40 | 61.72 | 35.20 | 42.66 | 46.53 | 55.19 |
| ColPali | 52.82 | 63.14 | 40.73 | 47.46 | 49.87 | 57.78 |
| LiLaC | 57.78 | 63.98 | 42.57 | 51.14 | 53.38 | 60.71 |
| IRCoT | 60.32 | 68.09 | 47.21 | 57.71 | 57.03 | 65.88 |
| FiF | **65.15** | **70.47** | **51.11** | **62.42** | **62.63** | **72.47** |

# 5. Experiments

## 5.1. Experimental Setups

**Datasets and evaluation metrics.** We evaluate open-domain multimodal *component* retrieval and downstream QA on three benchmarks: MultimodalQA (Talmor et al., 2021), MMCoQA (Li et al., 2022), and WebQA (Chang et al., 2022). Following LiLaC (Yun et al., 2025), we use the URL-annotated setting to reconstruct realistic webpage-style corpora, parsing each page into multimodal components (paragraphs, tables, images). This yields MultimodalQADoc (3,235 pages, avg. ∼37 components), MMCoQADoc (453 pages, avg. ∼32 components), and WebQADoc (7,662 pages, avg. ∼13 components). We defer their details to Appendix B.3. Consistent with prior work (Yun et al., 2025), we report retrieval Recall@k (R@k, $k \in \{1, 2, 5, 10\}$) and MRR@10: R@$k$ checks whether at least one ground-truth component appears in the top-$k$ list, and MRR@10 captures the rank of the first relevant component. For end-to-end QA, we feed the top-10 retrieved components into the same multimodal LLM and report Exact Match (EM) and token-level F1.

**Compared methods.** We compare FiF with strong baselines spanning graph traversal, agentic retrieval, and single-shot indexing. We include LiLaC (Yun et al., 2025), a layered-graph retriever designed for multi-hop scenarios, and IRCoT (Trivedi et al., 2023), an agentic retriever that interleaves retrieval with chain-of-thought reasoning. Since IRCoT was originally proposed for text-only corpora, we adapt it to our multimodal setting by (i) replacing its retriever with the same multimodal embedder used throughout our experiments and (ii) using the same multimodal LLM for reasoning and generation over multimodal components. We further compare with MoLoRAG (Wu et al., 2025c), a recent graph-based multimodal retriever

that augments embedding similarity with LLM-judged *logical relevance* during a fixed graph traversal. For *VisRAG* approaches, we employ VisRAG-Ret (Yu et al., 2024), which directly encodes document images via VLMs, and ColPali (Faysse et al., 2024), which uses late-interaction multi-vector embeddings from document images. We also compare with NV-Embed-v2 (Lee et al., 2024), a *TextRAG* baseline that embeds textualized components. We defer their details to Appendix B.4.

**Model configurations.** To ensure fair comparison, we standardize backbone models across all methods whenever applicable. We use MM-Embed (Lin et al., 2024) as the unified multimodal embedder, and the OpenAI API (GPT-5) (Singh et al., 2025) with reasoning_effort = low as the multimodal LLM for all planning, reasoning, reranking, and generation steps.

## 5.2. Retrieval Accuracy Comparison

We evaluate open-domain multimodal *component* retrieval on MultimodalQADoc, MMCoQADoc, and WebQADoc using Recall@$k$ ($k \in \{1, 2, 5, 10\}$) and MRR@10, with results reported in Table 1. FiF achieves the best performance across all three benchmarks and all reported cutoffs, indicating its overall effectiveness. Averaged across datasets, FiF reaches average Recall@10 of 80.26 and average MRR@10 of 83.16, improving over LiLaC +22.03% and +18.60%, respectively. Compared with the strongest agentic baselines IRCoT and MoLoRAG, FiF gains 12.88% and 11.39% Recall@10, and 9.31% and 8.92% MRR@10, respectively. The gap is even larger against single-shot embedding-based retrievers.

We analyze two interesting points. One is that the largest dataset-specific margin over LiLaC appears on WebQADoc at higher cutoffs, suggesting that WebQADoc more frequently requires *escaping* local neighborhoods to reach dispersed evidence. This aligns with WebQADoc's construction where ground-truth components are not necessarily adjacent or tightly coupled, making global navigation critical. In contrast, the performance gap over IRCoT is most pronounced on the more explicitly multi-hop benchmarks MultimodalQADoc and MMCoQADoc, where effectively leveraging the underlying link/structure signal (rather than pure global searching) is essential. Second, im-

*Table 3.* Efficiency (Time, # LLM Calls and API Usage) comparison of FIF and its competitors for the three benchmarks.

| Dataset | Algorithm | Time (ms) | | | | # LLM Calls | API Usage | | |
|---|---|---|---|---|---|---|---|---|---|
| | | Total | LLM | Vector Search | Embedding | | # Input Toks | # Output Toks | $ |
| MULTIMODALQA^Doc | VISRAG | 371 | 0 | 218 | 153 | 0.00 | 0 | 0 | 0.0000 |
| | COLPALI | 9,849 | 0 | 9,210 | 639 | 0.00 | 0 | 0 | 0.0000 |
| | LILAC | 19,528 | 15,943 | 3,153 | 432 | 1.00 | 1,165 | 1,119 | 0.0115 |
| | IRCOT | 95,744 | 95,140 | 119 | 484 | 5.44 | 41,943 | 3,296 | 0.0752 |
| | FIF | 114,591 | 113,642 | 538 | 411 | 7.92 | 48,863 | 4,785 | 0.1109 |
| MMCOQA^Doc | VISRAG | 362 | 0 | 215 | 147 | 0.00 | 0 | 0 | 0.0000 |
| | COLPALI | 1,836 | 0 | 1,173 | 663 | 0.00 | 0 | 0 | 0.0000 |
| | LILAC | 20,244 | 16,879 | 2,977 | 388 | 1.00 | 1,160 | 1,041 | 0.0107 |
| | IRCOT | 97,816 | 97,220 | 116 | 479 | 5.61 | 52,286 | 3,360 | 0.0753 |
| | FIF | 105,355 | 104,630 | 366 | 359 | 7.31 | 53,149 | 4,151 | 0.1037 |
| WEBQA^Doc | VISRAG | 386 | 0 | 225 | 161 | 0.00 | 0 | 0 | 0.0000 |
| | COLPALI | 7,919 | 0 | 7,298 | 621 | 0.00 | 0 | 0 | 0.0000 |
| | LILAC | 19,187 | 14,822 | 3,782 | 583 | 1.00 | 1,162 | 737 | 0.0077 |
| | IRCOT | 173,815 | 173,011 | 152 | 651 | 6.94 | 65,167 | 3,913 | 0.1082 |
| | FIF | 128,748 | 127,664 | 581 | 505 | 8.61 | 64,683 | 5,159 | 0.1278 |

*Table 4.* Ablation study analyzing retrieval accuracy and efficiency of different FIF variants on 20% subset of each dataset.

| Dataset | Variation | Retrieval Accuracy | | Efficiency | | | | |
|---|---|---|---|---|---|---|---|---|
| | | R@10 | MRR@10 | Time (ms) | # LLM Calls | # Input Toks | # Output Toks | $ |
| MULTIMODALQA^Doc | FAILURE IS FEEDBACK | 85.66 | 86.16 | 116,418 | 7.73 | 50,525 | 4,797 | 0.1083 |
| | w/o Backtracking Orchestration | 79.12 | 84.12 | 210,275 | 12.51 | 79,760 | 5,752 | 0.1468 |
| | w/o LLM Reasoning in traversal agent | 77.68 | 83.88 | 147,581 | 7.50 | 44,083 | 4,181 | 0.0872 |
| | w/o Global Hop | 75.59 | 82.41 | 87,991 | 7.19 | 53,622 | 4,146 | 0.1048 |
| | w/o Vector Granularity | 83.92 | 85.66 | 114,170 | 8.19 | 54,128 | 5,002 | 0.1173 |
| | w/o Subquery Planner | 76.21 | 83.81 | 67,791 | 5.17 | 34,861 | 3,382 | 0.0734 |
| WEBQA^Doc | FAILURE IS FEEDBACK | 80.92 | 88.80 | 124,772 | 8.52 | 63,860 | 5,153 | 0.1251 |
| | w/o Backtracking Orchestration | 77.61 | 87.81 | 230,292 | 13.94 | 90,197 | 7,402 | 0.1618 |
| | w/o LLM Reasoning in traversal agent | 78.52 | 88.24 | 134,202 | 7.49 | 51,575 | 4,191 | 0.0997 |
| | w/o Global Hop | 73.48 | 86.92 | 92,074 | 8.91 | 68,020 | 4,054 | 0.1231 |
| | w/o Vector Granularity | 79.11 | 88.02 | 133,945 | 8.90 | 67,990 | 5,511 | 0.1326 |
| | w/o Subquery Planner | 76.87 | 87.12 | 83,982 | 5.13 | 38,861 | 3,292 | 0.0791 |

*Table 5.* Retrieval accuracy of FIF across nine backbone LLMs on a representative 10% subset of each benchmark. Each cell is underlined when FIF with the corresponding backbone outperforms *both* GPT-5 competitors (IRCOT, MOLORAG) on that (dataset, metric); the best score in each column is in **bold**.

| Type | Backbone | MULTIMODALQA^Doc | | MMCOQA^Doc | | WEBQA^Doc | |
|---|---|---|---|---|---|---|---|
| | | R@10 | M@10 | R@10 | M@10 | R@10 | M@10 |
| Open | LLaMA 3.2 11B Vision | 68.13 | 72.43 | 61.38 | 56.91 | 72.79 | 84.62 |
| | Qwen3.5 27B | 74.54 | 82.63 | 65.18 | 63.45 | 72.82 | 83.84 |
| | GLM 4.6V (108B) | 76.43 | 81.43 | 68.94 | 66.17 | 78.05 | 87.88 |
| | KIMI K2.5 | 79.35 | 84.67 | 72.61 | 71.84 | 79.12 | 89.36 |
| Closed | GPT-4o-mini | 68.72 | 74.66 | 62.15 | 58.38 | 69.96 | 81.86 |
| | Claude Haiku 4.5 | 79.61 | 84.40 | 71.53 | 70.93 | 79.45 | 88.82 |
| | Gemini 2.5 Pro | 84.49 | 85.39 | 73.48 | 72.55 | 81.74 | 89.83 |
| | Claude Sonnet 4.6 | 85.14 | 86.81 | 74.25 | 73.62 | **82.06** | **91.57** |
| | GPT-5 | **85.82** | **86.82** | **74.91** | **74.88** | 81.21 | 89.24 |

provements are particularly strong at low-$k$, indicating that FIF not only increases *coverage* of relevant components but also ranks them substantially earlier.

## 5.3. End-to-end QA Accuracy Comparison

We measure end-to-end QA performance by feeding the top-10 retrieved components into the same multimodal LLM generator for every method, and report EM and token-level F1 in Table 2. FIF is consistently the best-performing method on all three datasets, achieving average EM/F1 of 59.63/68.45. Relative to LILAC, FIF improves EM by +16.37% and F1 by +16.79% on average, confirming that more reliable evidence discovery yields better grounded generation. Compared to IRCOT, FIF still provides a clear advantage of +8.71% EM and +7.14%

F1, despite both methods being agentic. Dataset-wise, the gains are especially visible on MMCOQA^Doc and WEBQA^Doc, consistent with Table 1 where FIF yields substantially higher top-$k$ retrieval accuracy.

## 5.4. Algorithm Efficiency

Table 3 reports wall-clock retrieval time (with breakdown into LLM, vector search, and embedding), the number of LLM calls, token usage, and estimated API cost. As expected, single-shot retrievers (VISRAG, COLPALI, NV-EMBED-V2) are the fastest and incur no LLM API cost during retrieval. Among agentic methods, FIF shows a slightly lower average runtime than IRCOT (116,231 vs. 122,458 ms), which is the strongest agentic baseline: while FIF executes more reasoning steps (7.31–8.61 vs. 5.44–6.94 LLM calls), it uses fewer input tokens and achieves comparable or lower latency overall. Notably, FIF is substantially faster on WEBQA^Doc (128,748 vs. 173,815 ms; −25.93%), suggesting that structure-aware navigation together with failure-aware re-anchoring reduces unproductive reasoning on large and sparsely connected corpora; The efficiency comes with a moderate increase in API usage compared to IRCOT ($0.10–$0.13 vs. $0.075–$0.108 per query), consistent with our higher retrieval/QA accuracy. Relative to LILAC, FIF is 5.20–6.71× slower in wall-clock time and incurs 9.64–16.60× higher API cost, quantifying the additional budget required by adaptive agentic control. We

view this as a favorable trade-off in settings where retrieval quality directly drives downstream correctness: against the strongest agentic baseline IRCoT, FIF attains comparable or lower latency——25.93% on WEBQA$^{Doc}$—while delivering +12.88% Recall@10 and +9.31% MRR@10 on average, and the resulting evidence translates into +8.71% EM and +7.14% F1 in downstream QA (Table 2).

### 5.5. Ablation Study

Table 4 isolates the contribution of each major design component in FIF on MULTIMODALQA$^{Doc}$ and WEBQA$^{Doc}$, reporting retrieval accuracy (R@10, MRR@10) alongside efficiency. We omit MMCoQA$^{Doc}$ from this analysis: because it is conversational, accumulated dialogue history can introduce confounding factors (e.g., varying context length and carryover information) that may blur the impact of individual retrieval modules. We run all ablations on a representative 20% subset of each dataset due to OpenAI API costs. (i) *History-aware backtracking orchestration.* Removing backtracking consistently hurts both *effectiveness* and *efficiency*: R@10 drops by 3.31–6.54, while runtime increases by ~80–85% and LLM calls rise by 62–64%. This highlights that backtracking prevents wasted hops by adapting effort only when needed. It also improves accuracy by returning to more promising anchors and narrowing candidates to the right neighborhood, rather than repeatedly exploring uninformative branches. (ii) *LLM reasoning inside the traversal agent.* Disabling LLM reasoning during traversal substantially degrades retrieval and can even *slow down* the search: on MULTIMODALQA$^{Doc}$, R@10 drops by 7.98 and runtime increases by 27%, despite lower per-step cost. We reason that additional replanning and extra iterations were triggered, as traversal is more likely to take ambiguous or unproductive hops without LLM reasoning. (iii) *Global hop.* Global hops are crucial for escaping local neighborhoods. When disabled, R@10 suffers the largest drop (10.07 on MULTIMODALQA$^{Doc}$; 7.44 on WEBQA$^{Doc}$), even though the variant becomes faster. This indicates that neighbor-based traversal alone is insufficient: some questions require jumping across distant regions of the corpus. (iv) *Adaptive vector-search granularity.* Removing granularity adaptation yields consistent but smaller drops (about 1.74–1.81 R@10) with minimal efficiency change. (v) *Subquery planner.* Removing replanning reduces LLM calls and cost substantially (e.g., −33% to −40% calls), but causes large recall drops (9.45 on MULTIMODALQA$^{Doc}$; 4.05 on WEBQA$^{Doc}$). Without the option to revise the plan, the Orchestrator is more likely to terminate early once progress stalls, which simultaneously lowers cost and recall.

### 5.6. Effect of the Backbone LLM

To verify that FIF's gains are attributable to its orchestration logic rather than to a particular backbone, we evaluate

FIF with nine LLM backbones of varying capability and openness on a representative 10% subset of each benchmark (consistent with the ablation protocol in §5.2). Table 5 reports R@10 and MRR@10 across the three datasets.

**(i) Gains are robust across capable LLMs.** From KIMI K2.5 and Claude Haiku 4.5 upward, FIF outperforms the strongest GPT-5 baselines (IRCoT, MOLORAG) on *every* (dataset, metric) cell, showing that gains stem from FIF's orchestration rather than a single proprietary backbone. **(ii) Operates reliably with open-weight backbones.** FIF's gains carry over to open weights: KIMI K2.5 already attains the (i) result, and mid-tier GLM 4.6V (108B) exceeds both GPT-5 competitors on MMCoQA$^{Doc}$ and WEBQA$^{Doc}$, trailing only on MULTIMODALQA$^{Doc}$ until the backbone is scaled up to KIMI K2.5. Below ~27B (LLaMA 3.2 11B Vision, GPT-4o-mini), accuracy drops visibly due to insufficient long-context reasoning over the serialized memory.

### 5.7. Additional Experiments

Additional experiments were conducted, but are detailed in the appendix due to space limitations. These include (i) a budget-matched comparison against IRCoT and MOLORAG at fixed per-query API cost (§ C.1), where FIF leads both baselines in the majority of (dataset, metric) cells; (ii) robustness to noisy or missing navigational edges (§ C.2), where removing 50% of navigational edges lowers R@10 by only 0.37–1.70; (iii) run-to-run stability under LLM stochasticity (§ C.3), with all standard deviations below 2%; and (iv) backtracking trigger frequency and re-anchoring success rates (§ C.4), where backtracking fires on 33–45% of the hardest queries yet still recovers 67–77% of them to perfect R@10.

### 6. Conclusion

We propose FAILURE IS FEEDBACK (FIF), an agentic multimodal retriever that models graph traversal as a *sequential decision process* over an explicit information state. FIF maintains a structured memory of subqueries and action history, transforming failures into actionable signals. Building on this, the Orchestrator makes retrieval *economically rational*: it dynamically chooses when to traverse, replan, or stop, and performs cost-aware strategy escalation—starting from low-cost matching and selectively pursuing accuracy over efficiency when ambiguity or failure justifies the added computation. Finally, FIF introduces history-aware backtracking via re-anchoring, using history to resume from more promising prior contexts. Extensive experiments on MULTIMODALQA, MMCoQA, and WEBQA demonstrate state-of-the-art accuracy on retrieval and downstream QA, validating the effectiveness of failure-aware traversal.

## Acknowledgements

This work was partly supported by Institute of Information & communications Technology Planning & Evaluation (IITP) grants funded by the Korea government (MSIT) (No. RS-2024-00509258, Global AI Frontier Lab, 50%; No. RS-2024-00454666, Developing a Vector DB for Long-Term Memory Storage of Hyperscale AI Models, 10%), the National Research Foundation of Korea (NRF) grant funded by the Korea government (MSIT) (No. RS-2025-00517736, 30%), and Basic Science Research Program through the National Research Foundation of Korea Ministry of Education (No. RS-2024-00415602, 10%).

## Impact Statement

FAILURE IS FEEDBACK (FiF) advances open-domain multimodal document retrieval, an enabling primitive for retrieval-augmented generation over webpages and PDFs. There are many potential societal consequences of our work, none of which we feel must be specifically highlighted here.

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

## A. On the Necessity of the MDP Formulation

A natural concern about the MDP framing in Section 4.1 is whether a Markov decision process formulation is necessary when our policy is a frozen LLM and there is no reward optimization or policy training. We clarify that the MDP serves an *architectural* role, not a learning-theoretic one.

**Structural role.** Although we do not optimize a reward, the MDP fixes the three structural ingredients—state $\mathcal{S}$, action set $\mathcal{A}$, and observation space—that the rest of FIF operates over. The state $s_t = M_t$ in particular encodes not only accumulated evidence but also the *explicit success/failure outcome* of every prior hop, recorded inside the action history $H_t$. Without this formal state, the agent would have to reason about retrieval progress purely through unstructured chat history, which is exactly the failure mode of prior agentic retrieval pipelines that we aim to address.

**Why the failure-conditioned mechanisms need this state.** History-aware backtracking (Section 4.3) needs a well-defined notion of "where the trajectory failed" and "which prior anchor to resume from"; both are direct readouts of $H_t$ inside $M_t$. Cost-aware strategy escalation similarly needs to consult the recorded outcome of the prior attempt on the same subquery to decide whether the next attempt should ascend the strategy ladder. The MDP formulation is what makes either signal well-typed and inspectable rather than implicit in the LLM's chat history; if we removed the formal state, both mechanisms would collapse back to a heuristic loop over unstructured prompts.

**Relation to standard agentic frameworks.** Framing OMDR as a sequential decision process under a unified $(\mathcal{S}, \mathcal{A}, \mathcal{T})$ also makes FIF directly comparable to standard agentic frameworks such as ReAct, which define planning–action–observation loops with explicit prompting rules and structured intermediate state but are similarly not characterized as learning-theoretic constructions. The MDP formulation therefore serves as the architectural substrate of FIF, with the policy-learning extension (e.g., reward-driven fine-tuning of the orchestrator over recorded $M_t$ trajectories) left as a clean future direction.

## B. Experimental Details

### B.1. Hardware and Software Settings

All experiments were conducted on a Linux server equipped with an Intel Xeon Gold 6230 CPU @ 2.10 GHz, 1 TB of RAM, and four NVIDIA RTX A6000 GPUs, running Ubuntu 22.04.3 LTS.

### B.2. Implementation Details

Our main hyperparameter is the vector-search shortlist size $k$ used by the Multi-strategy Traverser at both the document-level and component-level stages. Unless otherwise noted, we set $k=30$ for all experiments.

We also note implementation specifics of our released code that elaborate on—or, in a few places, deviate from—the abstractions used in the main text, so that the released code and the paper can be read side by side.

The initial hop at $t=0$ (where $\mathcal{D}_{anc} = \emptyset$) is implemented as a corpus-wide dense retrieval at the finest (subcomponent) granularity (Layer $V_2$); the top-matching subcomponents are then mapped up to their parent components through hierarchical edges to form the initial candidate set.

History-aware backtracking is carried out mainly by the TRAVERSE action. After each hop, the Traversal Evaluator not only records success/failure but also anticipates whether a failed hop is recoverable simply by escalating the strategy tuple. If it is, the Orchestrator immediately re-issues the same subquery from the current anchor with the next tuple on the cost ladder—escalating only $\tau_t$—as another TRAVERSE; otherwise it issues a TRAVERSE that performs the full re-anchoring (selecting a more promising prior anchor $\mathcal{D}_{anc}$), subquery revision ($q_t$), and strategy escalation ($\tau_t$). In our released code this evaluator-driven, escalation-only re-traversal is exposed as a lightweight `retry` shortcut.

### B.3. Benchmark Details

MULTIMODALQADoc: We use the extended version of `MultimodalQA`, following the augmentation procedure introduced in M3DocRAG (Cho et al., 2024). It spans diverse document modalities (text, images, and tables) and is designed to stress multi-hop reasoning over multiple documents. The evaluation split contains 2,441 questions grounded in 3235

webpages.

MMCoQADoc: We use an extended variant of MMCoQA that moves beyond the original distractor-only setting to evaluate conversational, multi-turn multimodal QA. The benchmark consists of coherent dialogue sessions in which later questions depend on earlier context and require aggregating evidence across text, images, and tables. It includes 5,753 questions grouped into 1,179 conversations, with a corpus of 218,285 text passages, 10,042 tables, and 57,058 images.

WEBQADoc: WEBQADoc is a Wikipedia-based multimodal QA benchmark with 4,966 questions over 7,662 documents. Because the original answers are often verbose, we rewrite them into concise references using the ChatGPT-5 OpenAI API with the prompt below.

```
Answer Concisification Prompt for WEBQA^Doc

You are an assistant that extracts concise answers from an Original Answer.

Task:
Given a Question, its Question Category (Qcate), and an Original Answer, extract a
    concise version of the answer.

Category hints:
- YesNo: respond only "Yes" or "No" matching the polarity of the Original Answer.
- text: return the minimal noun phrase/name that answers the question.
- choose: return only the chosen option or label.
- number: return the numeric value (and unit if present) without extra words.
- color: return the color term(s) only.
- shape: return the shape descriptor only.
- Others: follow the general concise rules below.

Rules:
- Output ONLY the concise answer text (no extra words, no labels, no punctuation-only
    output).
- Keep the minimum span that directly answers the Question.
- Prefer a single word when possible.
- If the question asks what object/thing, output the object noun phrase only (e.g.,
    fountain).
- If the question asks for a choice/comparison attribute (e.g., taller or shorter,
    happy or upset, or similar), output only the chosen option word from the answer (e.
    g., "taller", "upset").
- If the Original Answer is verbose by repeating or paraphrasing words/phrases already
    present in the Question, do NOT copy those repeated Question words into the concise
     answer; extract only the new, directly-question-answering information (If those
    repeated words are necessary for answering the question, then you may include them)
    .
- Preserve the original casing/pluralization as used in the Original Answer (e.g., "
    Circles").
- Do not include locations, explanations, or surrounding context unless they are
    required to uniquely answer the question.

Examples:
# Example 1
Question Category: YesNo
Question: Does a Minnetonka Rhododendron flower have petals in a cup shape?
Original Answer: No, a Minnetonka Rhododendron flower does not have petals in a cup
    shape.
Concise Answer: No

# Example 2
Question Category: Others
Question: What water-related object is sitting in front of the Torre del Reloj?
Original Answer: A fountain is sitting in front of the Torre del Reloj.
Concise Answer: fountain

# Example 3
```

```
Question Category: choose
Question: Is the fence in front of The Glass House in Fulham taller or shorter than a
    bicycle?
Original Answer: The fence in front of the building is taller than a typical bicycle.
Concise Answer: taller

# Example 4
Question Category: shape
Question: What shape is found 3 times on the front of the Archway in King Charles
    Street?
Original Answer: Circles may be spotted three times on the face of the Archway on King
    Charles Street.
Concise Answer: Circles

# Example 5
Question Category: choose
Question: Does the character in the work \"Beslotentuinfeest\" look happy or upset?
Original Answer: The character in the work \"Beslotentuinfeest\" looks upset.
Concise Answer: upset

# Example 6
Question Category: number
Question: How many more skis were used by Anders S\u00f6dergren during the 2010
    Olympics than were used by Martin Rulsch during the 2020 Winter Youth Olympics?
Original Answer: Anders S\u00f6dergren used two more skis during the 2010 Olympics.
Concise Answer: two

Inputs:
Question Category (Qcate): {qcate}
Question: {question}
Original Answer: {answer}
Concise Answer:
```

### B.4. Competitor Details

We provide a brief description of each competing method, ordered as in Table 1. They span single-shot embedding-based retrievers (NV-EMBED-V2, VISRAG, COLPALI), a structure-aware graph retriever (LILAC), and agentic retrievers (IRCOT, MOLORAG).

**NV-EMBED-V2** (Lee et al., 2024) is a *TextRAG* baseline. It first converts every multimodal component into text (e.g., captioning images and serializing tables), then embeds each textualized component into a dense vector using the NV-EMBED-V2 model, and retrieves by $k$-nearest-neighbor search over these vectors with respect to the embedded query. As a single-shot text retriever, it discards the native visual signal and performs no multihop traversal.

**VISRAG-RET** (Yu et al., 2024) is a vision-based, parsing-free retriever. Instead of textualizing the corpus, it renders each retrieval unit as an image and encodes it into a single dense vector with a vision-language encoder; at query time it embeds the query into the same space and retrieves by $k$-nearest-neighbor search over these image vectors. It thereby preserves the visual signal that text serialization would discard, but it represents each unit with one coarse whole-image embedding and retrieves in a single shot, without traversing the document graph or performing multihop search.

**COLPALI** (Faysse et al., 2024) is also a vision-based retriever, but it scores candidates through a COLBERT-style *late-interaction* mechanism rather than a single embedding. It encodes each unit image into a set of patch-level vectors and the query into token-level vectors, and scores a candidate by summing, over the query tokens, the maximum similarity (MaxSim) to that candidate's patches. This gives finer-grained query-to-region matching over visually rich content, yet retrieval is still a single-shot ranking over the flat set of units, with no structural or multihop reasoning.

**LILAC** (Yun et al., 2025) is the graph-based retriever most closely related to ours, and the one whose layered-graph formulation we build upon. It organizes the corpus as a layered component graph and, given a query, decomposes it into subqueries and runs an edge-wise beam search: at each step it expands along hierarchical and navigational edges and scores candidate nodes by late-interaction similarity to the current subquery, accumulating the components it visits. This makes

retrieval structure-aware and multihop, but it follows a fixed, similarity-driven expansion with a single uniform scoring rule, and it cannot revise its route or recover once a hop leads to a dead end.

**IRCoT** (Trivedi et al., 2023) is an agentic retriever that interleaves reasoning with retrieval. Starting from the query, it alternates between (i) generating the next chain-of-thought step from the query and the components retrieved so far, and (ii) using that reasoning step as a new query to retrieve additional components, repeating until it stops. Since it was proposed for text-only corpora over a flat index, we adapt it with the same multimodal embedder for retrieval and the same multimodal LLM for reasoning and generation. It retrieves over a flat component index without exploiting the document graph, and its only form of error correction is rewriting the query through further reasoning—it cannot backtrack to a prior context using recorded failures.

**MoLoRAG** (Wu et al., 2025c) is an agentic graph-based retriever. Over the corpus it builds a similarity graph that connects units whose embeddings are close, then lets a (vision-)language model traverse this graph to collect evidence: beginning from the most semantically similar units, it repeatedly examines their unvisited neighbors and scores each by combining embedding similarity with an LLM-judged *logical-relevance* score to the query, expanding under a fixed beam-style strategy with a preset exploration width and hop limit, and finally returns the top-scoring units. Since it was designed to retrieve pages within a single document, we apply it to our open-domain, component-level setting using the same multimodal embedder and LLM as the other methods. Unlike FIF, its traversal follows a single fixed strategy and it performs neither failure-conditioned backtracking nor cost-aware strategy escalation.

### B.5. Model Details

**Multimodal large language models:**

- `Open-AI ChatGPT5` (default backbone used in the main results).

We additionally evaluate FIF with the following backbones for our study across backbone LLMs reported in §5.6. All models are queried with their default decoding settings; for the OpenAI ChatGPT 5 API we set `reasoning_effort = low`, consistent with the main experiments. Smaller open-weight models (`LLaMA 3.2 11B Vision`, `Qwen3.5 27B`) are served locally on the hardware described in §B.1, while the larger open-weight models (`GLM 4.6V (108B)`, `KIMI K2.5`) are queried through the OpenRouter API.

- *Open-weight (served locally):* `LLaMA 3.2 11B Vision`, `Qwen3.5 27B`.
- *Open-weight (via OpenRouter API):* `GLM 4.6V (108B)`, `KIMI K2.5`.
- *Closed (proprietary API):* `GPT-4o-mini`, `Claude Haiku 4.5`, `Claude Sonnet 4.6`, `Gemini 2.5 Pro`, `GPT-5`.

**Text embedders**

- `NV-Embed-v2`: 7.85B parameters

**Cross-modal embedders:**

- `ColPali`: 3B parameters
- `VisRAG`: 3.43B parameters

**Multimodal embedders:**

- `MM-Embed`: 8.18B parameters

## C. Additional Experiments

### C.1. Budget-Matched Comparison

To separate the contribution of FIF's orchestration logic from the budget benefit of running more LLM calls, we cap the per-query API cost of IRCoT, MoLoRAG, and FIF at three identical limits ($0.05, $0.075, $0.10) and re-evaluate on a representative 10% subset with a fixed seed. Table 6 reports retrieval accuracy across all three datasets.

Two findings. (i) At equal per-query cost, FIF leads both baselines in the majority of (dataset, metric) cells, with clear

*Table 6.* Retrieval accuracy under varying per-query budget limits. All methods use the same 10% evaluation subset with a fixed seed.

| Algorithm | Budget | MULTIMODALQADoc | | | | | MMCoQADoc | | | | | WEBQADoc | | | | |
|---|---|---|---|---|---|---|---|---|---|---|---|---|---|---|---|---|
| | | R@1 | R@2 | R@5 | R@10 | M@10 | R@1 | R@2 | R@5 | R@10 | M@10 | R@1 | R@2 | R@5 | R@10 | M@10 |
| IRCoT | $0.05 | 37.23 | 54.04 | 66.69 | 71.27 | 77.18 | 42.88 | 52.87 | 61.98 | 62.14 | 62.11 | 36.01 | 53.22 | 63.97 | 69.32 | 79.96 |
| | $0.075 | 40.11 | 56.17 | 67.95 | 74.80 | 79.05 | 43.12 | 53.71 | 62.69 | 64.72 | 62.97 | 39.97 | 55.12 | 66.83 | 72.24 | 82.11 |
| | $0.10 | 40.95 | 57.20 | 70.23 | 75.30 | 79.51 | 43.25 | 54.28 | 63.39 | 65.22 | 63.18 | 40.32 | 55.95 | 68.53 | 73.10 | 82.91 |
| MoLoRAG | $0.05 | 38.11 | 51.22 | 62.93 | 68.66 | 79.27 | 43.12 | 54.23 | 61.13 | 63.76 | 62.78 | 38.17 | 53.39 | 64.29 | 68.03 | 80.72 |
| | $0.075 | 38.80 | 51.74 | 64.33 | 70.73 | 79.93 | 43.77 | 54.98 | 62.76 | 65.20 | 63.14 | 38.74 | 54.10 | 65.48 | 69.88 | 81.59 |
| | $0.10 | 39.80 | 52.61 | 66.19 | 72.79 | 80.30 | 43.91 | 55.12 | 63.41 | 66.18 | 63.72 | 39.01 | 54.53 | 65.72 | 70.23 | 82.01 |
| FiF | $0.05 | 38.71 | 55.19 | 66.18 | 72.02 | 77.40 | 43.10 | 54.81 | 61.97 | 62.96 | 63.62 | 42.94 | 57.00 | 66.74 | 70.10 | 86.73 |
| | $0.075 | 40.32 | 55.93 | 71.23 | 78.15 | 84.10 | 43.98 | 56.13 | 65.50 | 69.15 | 67.88 | 44.58 | 62.38 | 71.42 | 75.90 | 87.65 |
| | $0.10 | 41.43 | 56.91 | 73.14 | 83.12 | 85.20 | 45.13 | 57.91 | 68.13 | 74.11 | 73.91 | 45.57 | 63.37 | 73.23 | 78.31 | 88.09 |

dominance from the $0.075 budget upward. The advantage is most pronounced on WEBQADoc, where the corpus is sparsely connected and global routing is critical: even at the tightest $0.05 budget, FiF reaches 70.10 R@10 vs. 69.32 for IRCoT and 68.03 for MoLoRAG. At $0.05 the picture is more mixed—on MMCoQADoc a tighter budget favors the simpler baselines—but as the budget grows, FiF's lead widens consistently across all three datasets. (ii) FiF exhibits the highest budget elasticity, while MoLoRAG saturates. Going from $0.05 to $0.10, FiF's average R@10 across the three datasets rises by $+10.13$, compared with $+5.28$ for IRCoT and only $+3.61$ for MoLoRAG. FiF's orchestrator thus converts additional compute into useful reasoning steps (escalation, history-aware backtracking) rather than redundant retrieval, which cleanly disentangles the budget contribution from the orchestration logic.

## C.2. Robustness to Noisy or Missing Edges

To stress-test FiF in less-structured open-web environments, we randomly remove 50% of navigational edges on MULTIMODALQADoc and WEBQADoc and re-run retrieval. We note that these graphs are already inherently sparse: in the vanilla setting, only 53.0% (MULTIMODALQADoc) and 13.9% (WEBQADoc) of multi-hop queries have all gold documents connected through navigational paths; after edge removal, multi-hop connectivity drops further to 37.5% and 8.5% respectively.

*Table 7.* FiF's accuracy and efficiency under random 50% removal of navigational edges. Time is wall-clock in ms; "# LLM" is the average number of LLM calls per query; "$" is the average per-query API cost.

| Dataset | Setting | R@10 | M@10 | Time | # LLM | $ |
|---|---|---|---|---|---|---|
| MULTIMODALQADoc | Vanilla | 85.82 | 86.82 | 114,591 | 7.92 | 0.1109 |
| | Noisy ($-50\%$) | 84.12 | 85.19 | 121,415 | 8.84 | 0.1218 |
| WEBQADoc | Vanilla | 80.79 | 87.77 | 128,748 | 8.61 | 0.1278 |
| | Noisy ($-50\%$) | 80.42 | 87.16 | 132,570 | 9.11 | 0.1312 |

FiF degrades gracefully under heavy edge removal: R@10 drops by only 1.70 on MULTIMODALQADoc and 0.37 on WEBQADoc. Backtracking and global-hop escalation absorb the noise—scope is widened to recover alternative paths—at the modest cost of $+0.58$–0.92 additional LLM calls and $+0.003$–0.011 dollars per query. The smaller WEBQADoc degradation is expected: its vanilla connectivity is already only 13.9%, so FiF relies heavily on global hops even in the unmodified setting, making it less sensitive to further edge removal.

## C.3. Run-to-Run Stability

To assess sensitivity to LLM stochasticity, we sampled 5% of queries from each benchmark and ran FiF with the GPT-5 backbone 5 times per query under identical conditions, then computed the mean and standard deviation of the retrieval metrics over the 5 runs.

*Table 8.* Run-to-run stability of FiF (5 repeated runs on a 5% subset of each benchmark, GPT-5 backbone). All standard deviations are below 2.0%.

| Dataset | Stat. | R@1 | R@2 | R@5 | R@10 | M@10 |
|---|---|---|---|---|---|---|
| MULTIMODALQADoc | Avg. | 41.91 | 56.53 | 74.91 | 84.04 | 84.57 |
| | Stdev | 1.06 | 1.31 | 1.72 | 1.75 | 1.60 |
| MMCoQADoc | Avg. | 46.53 | 58.72 | 69.95 | 75.81 | 75.11 |
| | Stdev | 1.01 | 1.25 | 1.64 | 1.75 | 1.81 |
| WEBQADoc | Avg. | 42.61 | 62.57 | 74.09 | 80.05 | 87.05 |
| | Stdev | 0.86 | 1.23 | 1.55 | 1.71 | 2.04 |

Standard deviations remain below 2.0% across all metrics and benchmarks, which we attribute to the vector pre-filtering

step constraining downstream LLM reasoning to a consistent candidate set per query. For deployments requiring full determinism, our study across backbone LLMs (§ 5.6) confirms that FIF also operates effectively with open-weight backbones, which support deterministic decoding configurations.

## C.4. Backtracking Trigger Frequency and Re-anchoring Success

To assess how often history-aware backtracking fires in practice and how often the backtracked queries still recover perfect retrieval, we measure (a) the proportion of queries that trigger at least one backtracking event, and (b) the per-query *perfect-recall* rate (R@10 = 1.0) among backtracked vs. non-backtracked queries on each benchmark.

*Table 9.* Backtracking trigger frequency and per-query perfect-recall (R@10 = 1.0) rates among queries with vs. without at least one backtracking event.

| Metric | MULTIMODALQA$^{\text{Doc}}$ | MMCoQA$^{\text{Doc}}$ | WEBQA$^{\text{Doc}}$ |
|---|---|---|---|
| Queries with $\geq$ 1 backtracking | 38.2% | 33.1% | 45.3% |
| R@10 = 1.0 among backtracked queries | 76.6% | 71.4% | 67.3% |
| R@10 = 1.0 among non-backtracked queries | 86.7% | 88.2% | 80.1% |

We highlight two observations.

(i) Backtracking targets the inherently harder queries. Backtracking fires on 33.1–45.3% of queries; non-backtracked queries reach 80.1–88.2% perfect recall on average, whereas backtracked queries are substantially harder cases (dead ends, noisy edges, ambiguous hops) yet still attain 67.3–76.6% perfect recall.

(ii) The accuracy gap between backtracked and non-backtracked queries reflects selection, not failure. The $\sim$10–17 percentage-point gap is a property of which queries the mechanism activates on: precisely the ones that would otherwise remain at dead ends. That over two-thirds of these difficult queries still achieve perfect recall is direct evidence that history-aware re-anchoring effectively recovers from failure.

## D. Limitations

FIF incurs higher wall-clock latency and API cost than lightweight or fixed-call traversal methods such as LILAC. Although our cost-aware strategy escalation mitigates unnecessary computation, multi-step orchestration and occasional LLM-based reranking remain as bottlenecks. Practitioners under tighter latency or cost budgets can reduce per-query cost by selecting a cheaper backbone—our study across backbone LLMs (§ 5.6) shows that mid-tier backbones such as `Claude Haiku 4.5` and the open-weight `GLM 4.6V (108B)` retain most of FIF's accuracy benefit at a fraction of the cost. Our method assumes that each webpage/document is already parsed into a clean set of multimodal components and that navigational signals are reliably extracted. While we focus on URL-annotated corpora with structured navigation and component graphs, truly open-web deployment may involve noisier pages, weaker link signals, dynamic content, and heterogeneous layouts. An important direction for future work is to develop retrieval pipelines that are robust to diverse webpage structures and can jointly learn or adapt the parsing, graph construction, and retrieval policy so that the approach generalizes more seamlessly to arbitrary web content.

## E. Viable Future Work

### E.1. STOP and Unanswerable Queries

Our current STOP condition triggers when the Orchestrator judges that sufficient evidence has been collected or when the per-query tool-call budget is exhausted (set to 8, matching IRCoT). It does not explicitly detect *unanswerable* queries—cases where the relevant information does not exist in the corpus—and instead relies on the downstream generator to produce an "unanswerable" response when the accumulated memory $M_t$ remains insufficient. A principled extension is to monitor consecutive "not answerable" outputs from the Traversal Evaluator inside $H_t$: if escalations along the strategy ladder and history-aware backtracking still fail to produce a positive outcome, the Orchestrator can proactively trigger STOP before exhausting the budget. We leave a formal characterization of such early-termination conditions to future work.

### E.2. Interpretability and Controllability

Although the memory $M_t$ and the action history $H_t$ record the full decision trajectory, individual decisions—when to backtrack, which strategy tuple $\tau_t$ to escalate to—remain internal to the LLM-driven Orchestrator. A natural extension is

to require each agent to emit a short *rationale* alongside its decision; our prototype confirmed this improves diagnosability but increases per-hop latency, motivating a dual-mode design (a *debugging mode* with rationales for development versus a *deployment mode* without). For controllability, FIF's modular design already supports three levers: (i) restricting the strategy-tuple set to constrain the orchestrator's action space, (ii) injecting domain-specific rules into the Orchestrator prompt (e.g., per-corpus escalation preferences), and (iii) a human-in-the-loop mode in which users approve or override the Orchestrator's per-hop decision. A complete treatment of these levers—including their interaction with backtracking and budget—is left for future work.

