# OpenReview forum: "Failure is Feedback: History-Aware Backtracking for Agentic Traversal in Multimodal Graphs"
_ICML.cc/2026/Conference — ICML 2026 regular_

### Official Review · Reviewer_4pZg · 2026-03-09

**Soundness:** 2
**Presentation:** 3
**Significance:** 3
**Originality:** 2
**Overall Recommendation:** 4
**Confidence:** 3

**Summary:**

This paper studies open-domain multimodal document retrieval over interlinked corpora, aiming to retrieve evidence such as paragraphs, tables, and images through multi-hop traversal. The paper argues that prior graph-based methods are too rigid in practice, relying on hop-insensitive similarity scoring, fixed traversal procedures, and limited error recovery. To address this, the authors propose FAILURE IS FEEDBACK (FIF), which formulates retrieval as a sequential decision process over a layered multimodal graph. The system includes an LLM-based orchestrator that selects among traverse, plan, and stop actions; a multi-strategy traverser with adjustable scope, granularity, and reasoning depth; and a history-aware backtracking module that re-anchors retrieval based on prior failures. Experiments on three reconstructed multimodal QA benchmarks show consistent improvements in retrieval and downstream QA over LILAC, IRCOT, and several single-shot baselines.

**Compliance With Llm Reviewing Policy:**

Affirmed.

**Final Justification:**

Overall, I remain positive about this paper and support acceptance at the weak-accept level. The paper addresses an important problem in multimodal document retrieval, and the proposed framework is well motivated, readable, and empirically strong. My main concerns were about overstated novelty, heavy reliance on proprietary LLMs, limited ablation depth, and reproducibility details. The rebuttal addressed most of these concerns by providing stronger graph-agent baselines, budget-controlled comparisons, broader multi-LLM evidence, and clearer clarification of prompts and orchestration rules. I still think the core contribution is better viewed as an effective integration of agentic components rather than a deeply novel algorithmic advance, and some dependence on powerful underlying LLMs remains part of the story.

**Key Questions For Authors:**

1. How much of the improvement comes from better control logic, rather than from making more or stronger GPT-5 calls throughout the pipeline?

2. Why is the MDP formulation necessary here? There is no learned policy, reward optimization, or policy training, so the framing currently feels more descriptive than essential.

3. Did the authors compare against stronger graph-agent hybrid baselines, rather than mainly adapted text-oriented methods such as IRCOT?

4. If prior work is criticized for lacking a principled mechanism for deciding when expensive reasoning is needed, in what sense are the paper’s own cost ladder, escalation rules, and backtracking triggers more principled rather than simply better engineered heuristics?

5. Can the authors provide a stronger reproducibility package, including prompts, orchestration rules, trigger definitions, and a more controlled analysis of budget versus performance?

**Limitations:**

see weaknesses and questions.

**Strengths And Weaknesses:**

# Strengths

1. The paper gives a clear diagnosis of the problem: multimodal graph retrieval should not be reduced to fixed search under one similarity rule. The idea is not radically new, but it builds a meaningful agentic control layer over graph retrieval.

2. The empirical results look strong on the surface. FIF improves retrieval metrics across all three datasets and reported cutoffs.

3. The paper is generally readable. The motivation is clear, the figures are helpful, and the overall story is coherent: static traversal is brittle, while adaptive orchestration and failure-aware backtracking help.

4. The task is important. Multimodal retrieval over interlinked document graphs is a real bottleneck for multimodal RAG and multi-hop QA.

# Weaknesses

1. The novelty is somewhat overstated. Much of the method combines known ideas, including subquery planning, iterative retrieval, strategy escalation, LLM reranking, and memory-based control. The most distinctive part, history-aware backtracking, is still not developed deeply enough to feel like a principled algorithmic advance.

2. The evaluation is heavily entangled with proprietary LLM behavior. The same GPT-5 API is used for planning, reasoning, reranking, and generation, making it hard to separate gains from the orchestration logic itself versus repeated use of a strong closed model.

3. The baseline comparisons are not fully convincing. IRCOT is adapted from text-only retrieval to the multimodal setting, and it is unclear whether this adaptation is strong enough to be a truly competitive baseline.

4. The paper lacks enough detail on prompts, decision rules, and implementation specifics for high-confidence reproduction. Important behavior appears to depend on manually designed orchestration rules, including monotonic escalation, scope widening, failure-pattern definitions, and backtracking triggers.

5. The ablation study is limited. Running ablations on only 10 percent subsets is understandable given API cost, but it still weakens confidence. Since the modules are highly coupled, the current mostly single-drop analysis does not clearly show whether the gains come from backtracking, global hopping, planner flexibility, or simply more reranking opportunities.

6. The paper does not cleanly separate algorithmic benefit from budget benefit. FIF is often slower and more expensive than lighter baselines, and although it is sometimes faster than IRCOT, that does not establish broader cost-effectiveness. A fairer analysis would control for LLM calls, token budget, or API cost.

7. The paper criticizes prior work for lacking a principled mechanism for deciding when expensive reasoning is needed, but its own cost ladder, monotonic escalation, and backtracking triggers are also heuristic rule-based designs. That is fine in engineering terms, but the paper should present this more modestly.

8. A major concern is the heavy reliance on proprietary LLMs in both the system and evaluation pipeline. This weakens reproducibility and makes it unclear whether the gains come from the proposed method or from the strength of the underlying closed models. As written, the work reads more like a strong closed-model system demo than a broadly reproducible research contribution.

---

> ### Author Rebuttal · Authors · 2026-03-31
>
> We appreciate your constructive feedback and address all the concerns you raised. We further provide additional experiments, including a graph-agent baseline (MoLoRAG), 20% subset ablations, budget-controlled comparisons, and multi-LLM generalization. The tables are available at this [\<LINK\>](https://tinyurl.com/pzwm8nbn).
>
> **[W1: Novelty and Depth of History-Aware Backtracking]**
>
> As explicitly defined in our contributions in Section 1, the core novelty of FIF resides in the coupled integration of established agentic primitives into an empirically grounded control framework for open-domain multimodal graph traversal. Specifically, synthesize three structural mechanisms: (1) OMDR as a sequential decision process with explicit information states; (2) cost-aware strategy escalation triggered strictly upon recorded failure; and (3) history-aware backtracking via re-anchoring using failure traces.
>
> The algorithmic depth of our history-aware backtracking lies in the architectural decoupling of its trigger and execution. Our design ensures finite execution boundaries and system predictability, while leaving the re-anchoring process adaptive. Once triggered, the Orchestrator dynamically identifies where the trajectory failed, which prior anchor is most promising to resume from, and how to revise subsequent subqueries by explicitly conditioning on the structured failure trace. This synthesis preserves the predictable boundaries of a state machine while fully leveraging the LLM’s capability for context-aware trajectory recovery.
>
> **[W2/W8: Proprietary LLM Reliance]**
>
> To isolate our architectural contribution from proprietary LLM capacity, we evaluated FIF across 9 diverse LLMs (see Reviewer hshD). Crucially, FIF equipped solely with mid-tier open models strictly outperforms the strongest GPT-5-equipped competitors across all 3 benchmarks. This shows our gains stem from FIF’s orchestration logic, not proprietary model strength. Furthermore, open-weight deployment guarantees deterministic execution and reproducibility.
>
> **[W3/Q3: Graph-Agent Competitor]**
>
> To empirically validate our framework against graph-aware architectures, we evaluated against MoLoRAG (Wu et al., EMNLP 2025), a SOTA multimodal graph-agent pipeline. While MoLoRAG establishes a highly competitive baseline via LLM-driven scoring for unlabeled edges, FIF preserves a strict performance advantage over MoLoRAG across all datasets [Table A]. This confirms the empirical necessity of dynamic, history-aware traversal over static graph generation and fixed traversal strategies.
>
> **[W4/Q5:Reproducibility]**
>
> All prompt templates, orchestration rules, and source code are provided in Appendix D and supplementary materials. The manually designed rules are general and applied uniformly across all benchmarks without dataset-specific tuning.
>
> **[W5: Ablation Stability & Architectural Modularity]**
>
> To stress-test the statistical stability of our ablation, we expanded the subset scale from 10% to 20% [Table 2]. The empirical variance remained tightly bounded, and the relative module rankings remained invariant, confirming the stability of our initial conclusions. Regarding module coupling: FIF is intrinsically modular. Its components govern mutually exclusive stages of the retrieval sequence. Consequently, single-drop analysis cleanly isolates each module’s independent causal contribution without confounding effects, yielding the distinct, orthogonal accuracy-efficiency trade-offs reported. (Note: Restricting expensive ablations to representative subsets is standard practice in API-heavy LLM research at top venues).
>
> **[W6/Q1: Controlling Budget]**
>
> We conduct budget-control experiments (Table C) where all methods are capped at identical per-query API cost limits on 10% subsets. Key findings: (1) At equal budget, FIF consistently outperforms both baselines across all benchmarks. (2) FIF shows the highest budget sensitivity, while MoLoRAG exhibits saturating returns, indicating FIF's orchestration logic allocates additional budget more effectively.
>
> **[W7/Q4: Heuristic Nature of Manual Rules]**
>
> Despite the prompt-level constraints being heuristic, this does not make the overall framework heuristic: our core design—a memory-aware controller selecting actions conditioned on structured retrieval state—is a principled architectural choice similar to ReAct, which defines planning–action–observation loops with explicit prompting rules yet is not characterized as heuristic. In this study, we intentionally focus on the scope of the agentic framework design instead of policy learning.
>
> **[Q2: MDP Formulation Necessity]**
>
> Although we use a frozen LLM as a zero-shot policy without reward optimization, the MDP formulation is necessary to formally define the agentic behavior as sequential decision-making. Under a fixed policy and reward, our formulation can define the states, actions, and observations of our agent in a unified framework to understand our agent framework.

---

> > ### Author Rebuttal · Reviewer_4pZg · 2026-04-01
> >
> > Thank you for the detailed rebuttal. The response clarifies several important points and addresses almost all of my concerns.  However, one issues remain only partially resolved, regarding the claimed algorithmic novelty beyond a strong integration of existing agentic components, and the degree to which the reported gains can be disentangled from reliance on proprietary LLMs.

---

> > > ### Author Response · Authors · 2026-04-06
> > >
> > > Thank you for the detailed review and for leaning towards acceptance of our work. We address the two partially resolved points as follows:
> > >
> > > (1) **Novelty beyond integration.** We respectfully clarify why FIF's contribution extends beyond assembling known components. Our novelty is threefold, where the MDP formalization serves as the architectural foundation that structurally enables the other two:
> > >
> > > - MDP formalization of OMDR creates a structured information state ($M_t$​) that records accumulated evidence together with explicit success/failure outcomes, providing the structural substrate for the following two failure-conditioned mechanisms—both absent in all prior retrieval architectures.
> > >
> > > - History-aware backtracking is qualitatively different from standard state-reversion. When triggered, the Orchestrator performs three coordinated operations: (a) identifying where in the trajectory the failure occurred via explicit success/failure outcomes in $M_t$; (b) re-anchoring to the most promising prior successful context by scanning $H_t$; and (c) revising the subquery to incorporate constraints from the failure trace. No prior system—LILAC (fixed beam search), IRCoT (query rewriting without trajectory recovery), or MoLoRAG (static graph generation & LLM-guided fixed beam search)—possesses a structural mechanism for failure-conditioned trajectory recovery. The novelty further lies in decoupling trigger (deterministic rules ensuring finite execution) from execution (LLM-driven context-aware re-anchoring).
> > >
> > > - Cost-aware strategy escalation is structurally tied to the information state rather than being a static "try cheap first" heuristic: the Orchestrator reads explicit outcome records in $M_t$ and allocates reasoning compute per hop based on whether the previous attempt on the same subquery succeeded or failed.
> > >
> > > Critically, Table B of [\<LINK\>](https://anonymous.4open.science/r/ICML2026-FiF-tables-30B0/tables.pdf) (shown in our rebuttal) provides empirical evidence that these mechanisms are non-trivially synergistic: removing backtracking alone drops R@10 by 3.31–6.54 while simultaneously increasing runtime by ~81–85% and LLM calls by 62–64%. A single mechanism improving both accuracy and efficiency is a signature of principled architectural design that a simple combination of existing primitives would not produce. We will sharpen these distinctions in the camera-ready.
> > >
> > > (2) **Disentangling gains from proprietary LLMs.** We directly address whether the reported gains can be disentangled from reliance on proprietary LLMs. The key evidence is straightforward: FIF with the open-weight KIMI K2.5 (1T total / 32B active, MoE) consistently outperforms all three GPT-5-based baselines across every benchmark and metric (see table below). This demonstrates that FIF's improvements are attributable to its orchestration logic, not proprietary model strength.
> > >
> > > Even at a smaller scale, FIF with GLM-4.6V (106B total / 12B active) already surpasses GPT-5-based baselines on 4–5 out of 6 metrics, suggesting that the architectural contribution becomes dominant well before frontier-scale models are involved. We will include the full multi-LLM generalization results (9 backbone LLMs) in the appendix so that readers can verify this disentanglement directly.
> > >
> > > | Method | LLM Type | Backbone LLM | MMQA R@10 | MMQA MRR@10 | MMCoQA R@10 | MMCoQA MRR@10 | WebQA R@10 | WebQA MRR@10 |
> > > |---|---|---|---|---|---|---|---|---|
> > > | LILaC | Closed | GPT-5 | 72.23 | 79.12 | 60.86 | 53.36 | 64.23 | 77.87 |
> > > | IRCoT | Closed | GPT-5 | 74.97 | 82.24 | 64.62 | 62.84 | 73.72 | 83.15 |
> > > | MoLoRAG | Closed | GPT-5 | 77.82 | 83.54 | 67.18 | 63.18 | 71.15 | 82.32 |
> > > | FIF | Open | LLama 3.2 11B Vision | 68.13 | 72.43 | 61.38 | 56.91 | 76.79 | 84.62 |
> > > | FIF | Open | Qwen3.5 27B | 74.54 | 82.63 | 65.18 | 63.45 | 72.82 | 83.84 |
> > > | FIF | Open | GLM 4.6V (106B) | 76.43 | 81.43 | 68.94 | 66.17 | 78.05 | 87.88 |
> > > | FIF | Open | KIMI K2.5 | 79.35 | 84.67 | 72.61 | 71.84 | 79.12 | 89.36 |
> > > | FIF | Closed | GPT-4o-mini | 68.72 | 74.66 | 62.15 | 58.38 | 69.96 | 81.86 |
> > > | FIF | Closed | GPT-5 | 85.82 | 86.82 | 74.91 | 74.88 | 81.21 | 89.24 |
> > > | FIF | Closed | Gemini 2.5 Pro | 84.49 | 85.39 | 73.48 | 72.55 | 81.74 | 89.83 |
> > > | FIF | Closed | Claude Haiku 4.5 | 79.61 | 84.40 | 71.53 | 70.93 | 79.45 | 88.82 |
> > > | FIF | Closed | Claude Sonnet 4.6 | 85.14 | 86.81 | 74.25 | 73.62 | 82.06 | 91.57 |
> > >
> > > We sincerely appreciate your constructive feedback throughout this process.

---

### Official Review · Reviewer_5yt2 · 2026-03-11

**Soundness:** 4
**Presentation:** 4
**Significance:** 4
**Originality:** 3
**Overall Recommendation:** 5
**Confidence:** 3

**Summary:**

The paper introduces Failure is Feedback, an agentic framework for open-domain multimodal document retrieval. It conceptualizes graph traversal as a sequential decision process to overcome the limitations of rigid retrieval plans. The system employs a history-aware backtracking mechanism that utilizes the context of previous failures to re-anchor searches intelligently. Furthermore, it features an economically rational orchestrator that dynamically scales retrieval strategies from basic vector matching to intensive LLM reasoning based on current ambiguity and prior failures.

**Compliance With Llm Reviewing Policy:**

Affirmed.

**Key Questions For Authors:**

Could you elaborate on how the orchestrator decides to trigger a global hop versus a local neighbor search when the failure history is highly ambiguous?
How sensitive is the history-aware backtracking mechanism to missing or noisy edges in less structured open-web environments?
Are there specific heuristics or smaller language models that could be integrated into the orchestrator to reduce this latency for production use cases without losing the benefits of dynamic escalation?

**Limitations:**

yes

**Strengths And Weaknesses:**

Soundness is excellent because the submission is technically robust and supported by comprehensive empirical evidence. The experimental design is rigorous, comparing the proposed method against a solid mix of graph traversal, agentic retrieval, and single-shot baselines. The ablation studies effectively isolate the contributions of backtracking, LLM reasoning, and global hops, proving the necessity of each component. A minor weakness is the inherent latency and cost overhead of the agentic orchestration compared to single-shot methods, though the authors transparently acknowledge this tradeoff. Presentation is also excellent as the paper is well written and logically structured. The motivation for moving beyond uniform similarity metrics to a dynamic, history-aware approach is articulated clearly. The formalization of the retrieval task as an information-state Markov Decision Process is elegant and easy to follow.  Significance is excellent since addressing the brittleness of multihop retrieval in noisy, interconnected document graphs is a highly relevant problem. The introduction of cost-aware strategy escalation provides a practical blueprint for deploying LLM agents efficiently without sacrificing accuracy. This work will likely influence future retrieval-augmented generation architectures that operate over complex multimodal corpora. Originality is good. While agentic retrieval and layered component graphs exist, the novel combination of these ideas with explicit failure-driven feedback and dynamic cost escalation sets this work apart. Treating failed trajectories as constructive signals rather than mere dead ends is a creative and impactful perspective.

---

> ### Author Rebuttal · Authors · 2026-03-31
>
> We appreciate your constructive feedback. We have addressed all raised concerns with new experiments, including noisy-edge robustness analysis, multi-LLM generalization across nine backbone models, and detailed explanations of escalation and backtracking logic. We respond to each point below.
>
> **[W1: Latency and Cost Overhead]**
>
> Thank you for acknowledging our transparent discussion. As reported in Table 3, FIF's wall-clock latency is comparable to — and on WebQA, 25.93% faster than — the strongest agentic baseline IRCoT, despite achieving substantially higher accuracy. This is attributable to our cost-aware strategy escalation, which applies lightweight vector matching for straightforward hops and selectively engages LLM reasoning only when the hop is ambiguous or a prior attempt has failed. While FIF incurs higher latency and cost than single-shot methods, we believe there are practical scenarios where this tradeoff is favorable — particularly in applications where retrieval quality directly impacts downstream correctness, as evidenced by Table 2 (+8.71% EM, +7.14% F1 over IRCoT).
>
> **[Q3: Possible Orchestrator Heuristics for Production Deployment]**
>
> We address this from two angles.
>
> **Existing heuristics.** FIF’s orchestrator already incorporates cost-reducing mechanisms: (1) It constrains the strategy tuple space to a small set of meaningful combinations ordered by cost, expressed as an "escalation ladder" in the orchestrator prompt (Appendix D). This both eliminates less-effective configurations and provides a pre-defined ordering, reducing the orchestrator's reasoning burden. (2) Helper rules are explicitly specified in the orchestrator prompt to help it efficiently determine when to backtrack (e.g., "≥2 consecutive failures," "neighbor docs exhausted/irrelevant"), so the LLM can choose to follow pre-defined conditions rather than reasoning from scratch.
>
> **Cheaper backbone LLMs.** Practitioners can reduce per-query cost by selecting a capable but cheaper backbone while retaining FIF's core benefits. Our multi-LLM experiments show that mid-tier closed models like Claude Haiku 4.5 achieve flagship-competitive results at substantially lower cost, and some open-weight models onward surpass IRCoT with GPT-5 on all benchmarks. Please refer to Reviewer hshD's W1/Q2 for detailed results.
>
> **[Q1: Global Hop vs. Local Neighbor Search Under Ambiguous Failure History]**
>
> The decision follows the prompt-based escalation rules in Appendix D. There are two primary scenarios:
>
> **(i) New subtask with no viable anchor.** The orchestrator consults the action history for a suitable anchor. If no prior step's documents are relevant to the current subquery — or all candidates are associated with failures — it skips local search and directly initiates a global hop over the full corpus.
>
> **(ii) Escalation after local failure.** The orchestrator follows an escalation ladder (please  recall from Q3) from local to global scope. When local neighbor search fails at multiple granularities, it escalates directly to global retrieval rather than applying expensive LLM reasoning locally — reflecting the inductive bias that absent evidence cannot be recovered by stronger reasoning in the same neighborhood. When failure history is ambiguous (e.g., retrieved components are topically related but do not answer the subquery), the orchestrator favors a more thorough approach and proceeds with global search to maximize the chance of locating the ambiguous evidence.
>
> **[Q2: Sensitivity to Noisy/Missing Edges]**
>
> To empirically assess robustness, we simulated less structured environments by randomly removing 50% of navigational edges from both MMQA and WebQA. Please note that these graphs are already inherently noisy: in the vanilla setting, only 53.0% (MMQA) and 13.9% (WebQA) of multi-hop queries have all gold documents connected via navigational paths. After edge removal, connectivity among multi-hop queries dropped further to 37.5% and 8.5%, respectively.
>
> |Dataset|Setting|R@10|M@10|Time|#LLM Calls|$|
> |:-:|:-:|:-:|:-:|:-:|:-:|:-:|
> |MMQA|Vanilla|85.82|86.82|114,591|7.92|0.1109|
> |MMQA|Noisy (50% removed)|84.12|85.19|121,415|8.84|0.1218|
> |WebQA|Vanilla|80.79|87.77|128,748|8.61|0.1278|
> |WebQA|Noisy (50% removed)|80.42|87.16|132,570|9.11|0.1312|
>
> We confirmed that backtracking naturally detects that local traversals are insufficient and widens scope (local → global), recovering alternative paths. However, this recovery requires additional backtracking turns, increasing LLM calls by 0.58–0.92 per query and cost by $0.003–0.011. Since backtracking may not always be perfectly accurate, we observe modest R@10 drops of 1.70 (MMQA) and 0.37 (WebQA). The smaller degradation on WebQA is expected — its graph is already sparse (13.9% connected), so FIF already relies heavily on global hops in the vanilla setting, making it less sensitive to further edge removal.

---

> > ### Author Rebuttal · Reviewer_5yt2 · 2026-04-01
> >
> > Thank you for the thorough rebuttal. All three of my questions have been addressed convincingly.
> > The noisy-edge robustness analysis (Q2) is particularly compelling. Demonstrating that FiF degrades by only 1.70 R@10 on MMQA and 0.37 on WebQA under 50% edge removal — in graphs that are already quite sparse — provides strong evidence that the backtracking mechanism generalizes gracefully to less structured environments. The observation that WebQA's inherent sparsity makes it less sensitive to further edge removal is a clean and intuitive explanation.
> > The multi-LLM generalization results (shared from Reviewer hshD's response) substantially strengthen the paper. The finding that mid-tier open-weight models (GLM 4.6V and above) already surpass the strongest GPT-5 baseline confirms that the architectural contributions drive the gains rather than raw model capability. This was my primary residual concern regarding the latency/cost tradeoff (W1), and the availability of cheaper backbone options makes the practical deployment story much more convincing.
> > The clarification on global hop vs. local search escalation logic (Q1) is helpful and should be incorporated into the main text, as it makes the orchestrator's decision-making substantially more transparent to the reader.
> > I maintain my original recommendation of Accept. The new experiments on robustness, backbone generalization, and stability reinforce what was already a strong submission.

---

> > > ### Author Response · Authors · 2026-04-06
> > >
> > > Thank you for your strong support of our work from the outset and for the insightful questions that pushed us to strengthen the paper.
> > >
> > > We are glad that the additional experiments — particularly the noisy-edge robustness analysis and multi-LLM generalization — were convincing. As you suggest, we will incorporate the global-vs-local escalation logic into the main text in the camera-ready to improve transparency.
> > >
> > > We sincerely appreciate the constructive and encouraging review.

---

### Official Review · Reviewer_XerH · 2026-03-12

**Soundness:** 3
**Presentation:** 2
**Significance:** 3
**Originality:** 2
**Overall Recommendation:** 4
**Confidence:** 3

**Summary:**

This paper proposes a retrieval framework named FAILURE IS FEEDBACK (FiF) for open-domain multimodal document retrieval. FiF models the retrieval process as a sequential decision-making problem based on structured information states, employing a centralized "Orchestrator" to dynamically choose among "TRAVERSE," "PLAN," or "STOP" actions. Its core mechanisms include: (i) maintaining structured memory containing sub-queries and action history to transform retrieval failures into learnable feedback signals; (ii) introducing a cost-aware strategy escalation mechanism that dynamically balances low-cost matching and high-precision reasoning; and (iii) designing a history-aware backtracking mechanism that "re-anchors" to previously successful contexts to avoid repeating ineffective paths. Experimental results on three multimodal benchmarks—MultimodalQA, MMCOQA, and WEBQA—demonstrate that FiF significantly outperforms existing baselines (e.g., LILAC, IRCoT) in both component retrieval and end-to-end question answering tasks.

**Compliance With Llm Reviewing Policy:**

Affirmed.

**Key Questions For Authors:**

Extensibility to Unanswerable Questions.

Can the current STOP mechanism handle scenarios where the question has no corresponding answer (e.g., when the queried information does not exist in the corpus)? Have the authors considered extending the framework to support such cases? If so, what adjustments to the current design would be necessary?

**Limitations:**

Lack of Interpretability in Decision-Making.

Although structured memory is introduced to record decision trajectories, core decisions (e.g., when to backtrack, which strategy to choose) are still made by the LLM in a black-box manner. This makes diagnosing root causes and debugging the system exceptionally difficult when retrieval fails, limiting the method's interpretability and controllability.

**Strengths And Weaknesses:**

* Strength

1. The paper provides a clear formalization of the multimodal document retrieval problem, defining core concepts such as documents, components, navigational links, and layered component graphs.
2. The experimental design is comprehensive, covering three mainstream datasets and comparing against multiple types of baseline methods.
3. The ablation study is well-designed and systematically validates the effectiveness of FiF's core components.
4. The paper reports efficiency metrics such as latency, number of LLM calls, and API costs, providing an objective assessment of the method's practical usability.

* Weakness

1. The Orchestrator's decisions rely entirely on the LLM, but the paper does not discuss the stability of results across multiple runs of the same query. Due to the stochastic nature of LLM outputs, different runs may produce significantly different retrieval paths, affecting the reproducibility of results.
2. The STOP mechanism is designed solely based on whether sufficient evidence has been collected, failing to handle scenarios where the question itself has no corresponding answer (e.g., unanswerable questions). This limits the framework's extensibility to broader application scenarios.
3. Some symbols and content are not adequately explained, such as the meaning of variables in formulas (e.g., "x") and the initialization method for anchor document sets.
4. The explanation of the backtracking mechanism lacks clarity; readers need to consult the prompt templates in the appendix to understand its specific implementation, which compromises the readability of the main text.
5. The core components (e.g., graph structure and agent paradigm) originate from existing research, and the overall framework's originality primarily lies in combination and adaptation rather than proposing entirely new theories or paradigms.

---

> ### Author Rebuttal · Authors · 2026-03-31
>
> We appreciate your constructive feedback. We have addressed all concerns with new experiments (run-to-run stability, 9-LLM generalization) and revised descriptions for backtracking and notation. We respond to each point below.
>
> **[W5: The Locus of Originality: Principled Architectural Synthesis]**
>
> We position FIF not as the invention of isolated algorithmic primitives, but as the first principled architectural synthesis for agentic traversal over open-domain multimodal graphs. While discrete building blocks (e.g., subquery planning, iterative retrieval) exist, they lacked the structural connective tissue to operate over noisy multimodal evidence.
>
> The core scientific originality lies in formalizing OMDR as a finite-horizon MDP, which structurally binds isolated mechanisms into a cohesive state machine. By defining the information state, action space, and transition dynamics, this formulation intrinsically enables two novel mechanisms: (1) history-aware backtracking that transcends standard state-reversion by constructively converting failed traversals into dynamic re-anchoring signals—jointly revising the physical anchor and semantic subquery conditioned strictly on the explicit failure trace; and (2) cost-aware strategy escalation that structurally allocates reasoning compute per hop based on explicitly recorded outcomes in $M_t$. These dynamic, failure-conditioned mechanisms represent a fundamental paradigm shift from prior flat-index or static-plan retrieval architectures, which do not possess structural counterparts for trajectory recovery.
>
> **[W1: Stability and Reproducibility]**
>
> We verified reproducibility from two angles.
> (i) We confirmed that FIF generalizes over open-weight models producing consistent results aligned with model capability — please refer to Reviewer hshD’s W1/Q2 for detailed results across nine LLMs. Notably, open-weight models support deterministic decoding, which guarantees full reproducibility in deployment.
> (ii) To directly assess run-to-run stability with GPT-5, we sampled 5% of queries from each dataset and ran FIF 5 times per query under identical conditions:
>
> |Dataset|Stat|R@2|R@5|R@10|MRR@10|
> |:-|:-:|:-:|:-:|:-:|:-:|
> |MultimodalQA|Avg|56.53|74.91|84.04|84.57|
> ||Stdev|1.31|1.72|1.75|1.60|
> |MMCoQA|Avg|58.72|69.95|75.81|75.11|
> ||Stdev|1.25|1.64|1.75|1.81|
> |WebQA|Avg|62.57|74.09|80.05|87.05|
> ||Stdev|1.23|1.55|1.71|1.94|
>
> Standard deviations remain below 2.0%, due to vector pre-filtering that constrains LLM reasoning to consistent candidate sets.
>
> **[W2/Q1: Extensibility to Unanswerable Questions]**
>
> We acknowledge that the current STOP condition is not explicitly designed to detect unanswerable queries early. However, in practice, FIF mitigates this by imposing a maximum tool call limit (set to 8, consistent with IRCoT); when exhausted without sufficient evidence, the downstream generator can naturally produce an “unanswerable” response based on the memory. A principled future extension would monitor consecutive failures in $H_t$: if the Traversal Evaluator consistently returns “not answerable” despite escalations, the Orchestrator could proactively trigger STOP before exhausting the budget.
>
> **[W3: Symbol and Initialization Clarification]**
>
> We apologize for the insufficient notation.
>
> (i) The notation $x(u)$ was insufficiently defined — we will clarify that $x(\cdot)$ maps a node to its embedding vector.
>
> (ii) Regarding anchor initialization, the anchor document set $\mathcal{D}_{\text{anc}}$ is initialized to $\emptyset$ at the very first hop ($t=0$).
>
> **[W4: Clarity of the Backtracking Mechanism]**
>
> We will expand Section 4.3 with the following description:
>
> When a traversal fails, the Orchestrator first retries with a more thorough strategy. When consecutive failures occur within the history — or the highest-cost strategy has been attempted and still failed — the Orchestrator triggers backtracking through three coordinated operations:
>
> (1) Revise the subquery $q_t$ to incorporate constraints learned from failures;
> (2) Re-anchor by scanning $H_t$ for the most recent successful record whose evidence best matches the revised subquery, resetting $\mathcal{D}_{\mathrm{anc}}$.
> (3) Escalate the strategy $\tau_t$ to avoid repeating the failed traversal behavior. In practice, valid tuples are constrained to a fixed escalation ladder to reduce the orchestrator’s reasoning burden.
>
> **[L1: Interpretability and Controllability]**
>
> For interpretability, a natural extension is requiring each agent to generate rationales alongside decisions. Our early prototype included this, but it increases latency; a practical solution is a dual-mode design (debugging mode with rationales vs. deployment mode without). For controllability, FIF’s modular design already supports: (i) constraining strategy tuples to restrict the action space, (ii) injecting domain-specific rules into the orchestrator prompt, and (iii) human-in-the-loop mode where users approve or override decisions at each hop.

---

> > ### Author Rebuttal · Reviewer_XerH · 2026-04-08
> >
> > Thanks to the authors' responses. As all the reviewers share the consistent opinion, I will maintain my score.

---

> > > ### Author Response · Authors · 2026-04-08
> > >
> > > Thank you for acknowledging that all concerns have been fully resolved, and for the constructive feedback that meaningfully improved our paper — regarding run-to-run stability (W1), the STOP mechanism's extensibility to unanswerable questions (W2/Q1), symbol and initialization clarification (W3), backtracking clarity (W4), the discussion on originality (W5), and interpretability/controllability (L1). We will incorporate the expanded backtracking description, symbol clarifications, and other revisions into the camera-ready version.
> > >
> > > We sincerely appreciate your constructive feedback throughout this process.

---

### Official Review · Reviewer_hshD · 2026-03-12

**Soundness:** 3
**Presentation:** 3
**Significance:** 3
**Originality:** 3
**Overall Recommendation:** 4
**Confidence:** 2

**Summary:**

This paper addresses open-domain multimodal document retrieval (OMDR), given a query, the system find a small ranked set of relevant components from this large, noisy, and interlinked graph. The authors proposed FAILURE IS FEED-BACK (FIF), an LLM-driven agentic retriever that which casts subgraph retrieval as a sequential decision process. It features: (1) history-aware backtracking mechanism, (2) economically-rational agentic workflow, and (3) dynamic subquery planning based on accumulated evidence. Extensive experiments show that FIF achieves state-of-the-art retrieval on three multimodal QA benchmarks.

**Compliance With Llm Reviewing Policy:**

Affirmed.

**Key Questions For Authors:**

1. How often does backtracking actually trigger? What is the success rate of re-anchored traversals?
2. Does FiF work with cheaper LLMs like GPT-4o-mini, Llama-3, or Claude? The framework relies on GPT-5 for orchestration, evaluation, planning, and reranking. What's the minimum LLM capability needed?

**Limitations:**

See weekness above

**Strengths And Weaknesses:**

Strengths

- Strong empirical results: FiF achieves the best performance across all three benchmarks on both retrieval (R@k, MRR@10) and end-to-end QA (EM, F1) metrics, with substantial improvements.

- Comprehensive ablation study: Extensive ablation studies (Table 4) validate each component's contribution.

- Practical efficiency analysis: Table 3 provides detailed wall-clock time breakdowns, LLM call counts, token usage, and dollar costs for actionable deployment insights.

Weakness
- All results use GPT-5. The lack of experiments with other backbone LLMs prevents assessing the framework's robustness to model choice and generalization capability.

---

> ### Author Rebuttal · Authors · 2026-03-31
>
> We appreciate your constructive feedback. We have addressed all raised concerns with new experiments, including multi-LLM generalization across nine backbone models and a detailed analysis of backtracking trigger frequency and re-anchoring success rates. We respond to each point below.
>
> **[W1/Q2: Backbone LLM Generalization & Capacity-Dependent Performance Trends]**
>
> Thank you for this important concern. We fully agree that demonstrating robustness across different backbone LLMs strengthens the generalizability claim of our framework. To address this, we conducted additional experiments using nine LLMs of varying capability and cost on all three benchmarks. Due to API costs, we evaluate on representative 10% subsets of each dataset, consistent with our ablation setup (§5.5).
>
> |Model Type|Backbone LLM|MultimodalQA R@10|MultimodalQA MRR@10|MMCoQA R@10|MMCoQA MRR@10|WebQA R@10|WebQA MRR@10|
> |:-:|:-:|:-:|:-:|:-:|:-:|:-:|:-:|
> |Open|LLama 3.2 11B Vision|68.13|72.43|61.38|56.91|76.79|84.62|
> |Open|Qwen3.5 27B|74.54|82.63|65.18|63.45|72.82|83.84|
> |Open|GLM 4.6V|76.43|81.43|68.94|66.17|78.05|87.88|
> |Open|KIMI K2.5|79.35|84.67|72.61|71.84|79.12|89.36|
> |Closed|OpenAI GPT-4o-mini|68.72|74.66|62.15|58.38|69.96|81.86|
> |Closed|OpenAI ChatGPT 5|**85.82**|**86.82**|**74.91**|**74.88**|81.21|89.24|
> |Closed|Gemini 2.5 Pro|84.49|85.39|73.48|72.55|81.74|89.83|
> |Closed|Claude Haiku 4.5|79.61|84.4|71.53|70.93|79.45|88.82|
> |Closed|Claude Sonnet 4.6|85.14|86.81|74.25|73.62|**82.06**|**91.57**|
>
> We highlight two findings:
>
> **(i) FIF's gains are robust across capable LLMs.** From KIMI K2.5 / Claude Haiku 4.5 and above, FIF consistently outperforms the strongest baseline IRCoT *using GPT-5* across all three benchmarks (IRCoT R@10/M@10: 74.97/82.24 on MMQA, 64.62/62.84 on MMCOQA, 73.72/83.15 on WebQA). This shows that FIF's core contributions — history-aware backtracking, cost-aware escalation, and agentic orchestration — generalize beyond a single backbone.
>
> **(ii) FIF operates reliably with mid-tier LLMs starting from ~27B–90B scale, including open-weight models.** Starting from GLM 4.6V (108B, open-source), FIF surpasses IRCoT with GPT-5 on all three datasets, demonstrating that practitioners can deploy FIF effectively without relying on proprietary APIs. Scaling further reveals a clear capability hierarchy: KIMI K2.5 ≈ Claude Haiku 4.5 < ChatGPT 5 ≈ Gemini 2.5 Pro ≈ Claude Sonnet 4.6, where the top-tier group achieves near-parity with each other while maintaining substantial margins over lower-tier models. Accuracy degrades substantially below the ~27B threshold (GPT-4o-mini, Llama 3.2 11B): we attribute this to insufficient reasoning capacity over the action history, which averages 48–64K total input tokens per query with an average of 7.5K tokens per orchestrator call, demanding robust long-context comprehension.
>
> We will include these results in the revised manuscript.
>
> **[Q1: Backtracking Trigger Frequency & Re-anchoring Success Rate]**
>
> Thank you for this question. We analyzed backtracking behavior on all three datasets, measuring (1) the proportion of queries triggering at least one backtracking event, and (2) the success rate (Recall@10 = 1.0) among backtracked queries.
>
> |Metric|MultimodalQA|MMCoQA|WebQA|
> |:-|:-:|:-:|:-:|
> |Queries with ≥1 backtracking|38.2%|33.1%|45.3%|
> |Recall@10 = 1.0 rate among backtracked queries|76.6%|71.4%|67.3%|
> |Recall@10 = 1.0 rate among non-backtracked queries|86.7%|88.2%|80.1%|
>
> We highlight two key observations:
>
> **(i) Backtracking targets inherently harder queries.** Backtracking triggers on 38.9% of queries on average, selectively activating on difficult cases where initial traversal fails. Non-backtracked queries achieve 85.0% perfect recall on average, whereas backtracked queries — despite being substantially harder — still achieve 71.8%. Without the backtracking mechanism, these queries would remain at dead ends.
>
> **The gap between backtracked and non-backtracked recall (85.0% vs. 71.8%) reflects the selective nature of the mechanism.** Backtracking activates precisely on the hardest cases — queries where the retriever encounters dead ends, noisy edges, or ambiguous hops. The fact that over two-thirds of these difficult queries still achieve perfect recall demonstrates that history-aware re-anchoring effectively recovers from failures.
>
> We will include this analysis in the revised manuscript.

---

### Decision · Program_Chairs · 2026-04-30

**Decision:**

Accept (regular)

**Comment:**

This paper introduces the "Failure is Feedback" framework for open-domain multimodal document retrieval. The idea of using graph is nice, allowing the system to naturally cast retrieval as a sequential decision process equipped with history-aware backtracking and cost-aware strategy escalation. Reviewers unanimously appreciated the strong empirical results across three distinct benchmarks, the thorough ablation studies, and the transparent efficiency analysis. Although initial reviews raised valid concerns regarding the framework's reliance on proprietary LLMs and baseline comparisons, the authors provided a convincing rebuttal that successfully resolved these issues by demonstrating robust performance using open-weight models and in noisy-edge scenarios.